## Registered report

health and disease and epidemiology/psychology

COVID-19 pandemic, adolescence, mental health, depression, externalizing, wellbeing

**Author for correspondence:**
Praveetha Patalay
e-mail: p.patalay@ucl.ac.uk

†joint first.
‡joint last.

# The impact of the COVID-19 pandemic on adolescent mental health: a natural experiment

Rosie Mansfield[1,†], Joao Santos[3,†], Jessica Deighton[4], Daniel Hayes[4,5], Tjasa Velikonja[4], Jan R. Boehnke[6,‡] and Praveetha Patalay[1,2,‡]

[1]Centre for Longitudinal Studies, Social Research Institute and [2]MRC Unit for Lifelong Health and Ageing, Population Science and Experimental Medicine, University College London, London, UK
[3]Manchester Institute of Education, University of Manchester, Manchester, UK
[4]Evidence Based Practice Unit, UCL and Anna Freud National Centre for Children, and Families, London, UK
[5]Health Service and Population Research, Institute of Psychiatry, Psychology and Neuroscience, King's College London, London, UK
[6]School of Health Sciences, University of Dundee, Dundee, UK

RM, 0000-0002-8703-5606; JS, 0000-0002-8212-3025; JRB, 0000-0003-0249-1870; PP, 0000-0002-5341-3461

Despite widespread concern about the impact of COVID-19 on adolescent mental health, there remains limited empirical evidence that can causally attribute changes to the pandemic. The current study aimed to overcome existing methodological limitations by exploiting a serendipitously occurring natural experiment within two ongoing, multi-phase cluster randomized controlled trials. Depressive symptoms (primary outcome), externalizing difficulties and life satisfaction (secondary outcomes) were assessed at baseline (phase 1 [pre-COVID-19 group]: September – October 2018, phase 2 [COVID-19 group]: September – October 2019) and 1-year follow-up (pre-COVID-19 group: January – March 2020, COVID-19 group: February – April 2021). Participants in phase 1 ($N = 6419$) acted as controls. In phase 2, participants ($N = 5031$) were exposed to the COVID-19 pandemic between the baseline and follow-up assessments providing a natural experimental design. The primary analysis used a random intercept linear multivariable regression model with phase (exposure to the COVID-19 pandemic) included as the key predictor while controlling for baseline scores and individual and school-level covariates. Depressive symptoms were higher and life satisfaction scores lower in the group exposed to the COVID-19 pandemic. Had the COVID-19 pandemic not occurred, we

estimate that there would be 6% fewer adolescents with high depressive symptoms. No effect of exposure to the pandemic on externalizing difficulties was found. Exploratory analyses to examine subgroup differences in impacts suggest that the negative impact of the COVID-19 pandemic on adolescent mental health may have been greater for females than males. Given the widespread concern over rising adolescent mental health difficulties prior to the pandemic, this paper quantifies the additional impacts of the pandemic. A properly resourced, multi-level, multi-sector public health approach for improving adolescent mental health is necessary. Following in-principle acceptance, the approved Stage 1 version of this manuscript was preregistered on the OSF at https://doi.org/10.17605/OSF.IO/B25DH. This preregistration was performed prior to data analysis.

# 1. Introduction

## 1.1. Background

Prior to the COVID-19 pandemic, there was widespread concern over rising mental health difficulties experienced by adolescents. In 2017, between 14 and 17% of adolescents aged 11–19 were found to meet diagnostic criteria for at least one mental health disorder in England [1]. Cross-cohort studies have demonstrated increases in internalizing difficulties that indicate a deterioration of adolescent mental health over time [2,3]. It is important to understand whether the COVID-19 pandemic has contributed further to increased mental health difficulties in adolescence.

Despite widespread concern and media coverage about the impact of COVID-19 and related school closures on adolescent mental health [4], there remains limited robust empirical evidence that can causally attribute mental health changes to the pandemic [5,6]. To isolate the pandemic's effect, studies must include pre-pandemic assessments of symptoms [7] and account for age effects given known developmental patterns in mental health difficulties [8]. Even when longitudinal data are available, results must be considered in the context of secular trends in child and adolescent mental health [9]. Differentiating between age or developmental changes and the impact of the COVID-19 pandemic is of particular relevance for younger populations, as internalizing symptoms are known to increase year-on-year from mid-adolescence [10,11].

Much of the existing evidence is based on cross-sectional studies with no pre-pandemic assessments of mental health. Longitudinal data for this population are limited and in this age group pose the unique challenge of differentiating between developmental change and COVID-19 impact. A living systematic literature review investigating the changes in mental health symptoms within the same individuals from pre-COVID-19 and across distinct phases of the pandemic identified only four studies with child and adolescent samples (by June 2021 when this was written), none of which were from the United Kingdom (UK) [12]. Findings from these few studies are mixed, with increased internalizing symptoms reported in Australia [13] and increased conduct and overall difficulties reported in Spain [14]. By contrast, fewer depressive and externalizing symptoms were reported in China [15] and the Netherlands, respectively [8]. More recently, results from a longitudinal, population-based study in Iceland revealed trajectories of pre-pandemic depressive symptoms between 2016 and 2018 and during the COVID-19 pandemic [16]. Adolescents aged 13–18 years reported significantly more depressive symptoms during the pandemic, and mental well-being decreased beyond what might be expected based on existing time trends of adolescent mental health [6].

In the UK, data from an ongoing regional cohort (Wirral Child Health and Development Study) revealed stark increases in young adolescents' depressive symptoms, post-traumatic stress disorder and externalizing difficulties during the COVID-19 pandemic [17]. At a national level, the only population-based data indicating changes in mental health has come from the COVID-19 follow-up of the 2017 prevalence study [18]. A higher proportion of children were found to be experiencing mental health difficulties, albeit methodological limitations around low response rates and differences in the mode and method of assessment before and during the pandemic [19]. Both these UK studies are limited in their ability to separate developmental changes from pandemic-related impact.

## 1.2. Objectives

To address some of these methodological challenges, this study exploited the serendipitous design of two large, ongoing multi-phase intervention trials. Using two cohorts of students (figure 1), we are better able

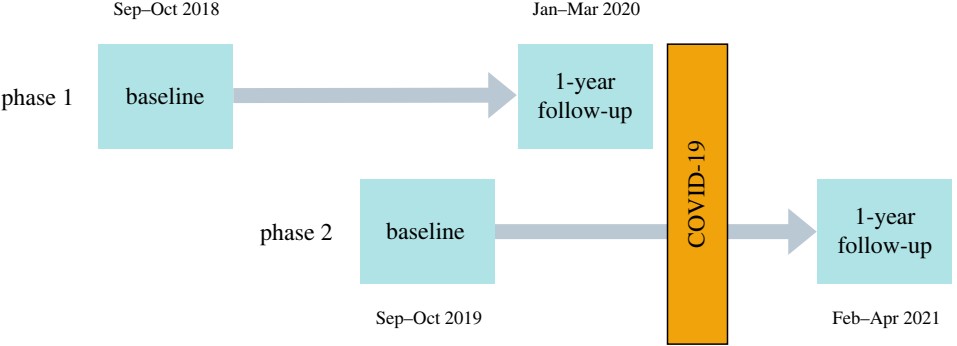

**Figure 1.** Study design: process chart with timelines and assessments in each phase.

to isolate the impact of the COVID-19 pandemic on adolescent mental health from age and longer-term trends than previous studies. In each study phase, baseline measures were assessed with adolescents aged 11–15 years across secondary schools. While in both phases the baseline assessment happened prior to the pandemic, the follow-up in phase 1 (pre-COVID-19 group) was assessed just before the pandemic (January–March 2020). Adolescents participating in phase 2 (COVID-19 group) experienced the pandemic and school closures between baseline and follow-up (February–April 2021). Hence, this paper aimed to answer the following research question: What impact has the COVID-19 pandemic had on adolescent mental health, specifically, depressive symptoms (primary outcome), externalizing difficulties, and life satisfaction (secondary outcomes)? We hypothesized that after controlling for baseline variables, levels of depressive symptoms and externalizing difficulties would be higher, and life satisfaction lower, during the COVID-19 pandemic compared to before.

There has been some evidence from UK studies tracking families throughout the pandemic, that children with special educational needs (SEN) and from low-income homes were particularly impacted by COVID-19 related school closures and lockdown [20]. To investigate whether this was the case in our study population, we subsequently examined whether there were socio-demographic differences (based on gender, ethnicity, socio-economic disadvantage and special educational needs) in the impact of the COVID-19 pandemic on adolescent mental health outcomes.

# 2. Material and methods

## 2.1. Study design and setting

Details of the trials from which these data are drawn are described below. The Education for Wellbeing Programme (EfW) is an evaluation of five school-based, mental health and wellbeing interventions which are organized into two parallel-group cluster randomized controlled trials (RCTs) [21,22]. Interventions in Schools for Promoting Mental Wellbeing: Research in Education (INSPIRE), is a four-arm cluster RCT comparing three interventions (Mindfulness, Relaxation and Strategies for Safety and Wellbeing) to usual provision (control). Approaches for Wellbeing and Mental Health Literacy: Research in Education (AWARE) is a cluster RCT consisting of three arms, comparing two mental health education interventions (Youth Aware of Mental Health (YAM) and The Guide) to usual provision (control). Randomization of schools was conducted following baseline data collection by King's Clinical Trials Unit using an equal allocation ratio [1 : 1 : 1]. Minimization was applied for deprivation (free school meal eligibility), geographical region (London, Greater Manchester and Northwest, Bath & Bristol and Durham), urban/rural location, and mental health provision reported at baseline (prior interventions coded absent/present) to ensure that conditions were comparable. For a full description of interventions, study design and measures, see the trial protocol papers [21,22].

Due to the size of the two trials, schools were recruited in two phases (figure 1; allocation to all interventions in both phases). Outcomes were assessed at baseline (prior to intervention randomisation) (phase 1 [pre-COVID-19 group]: September – October 2018, phase 2 [COVID-19 group]: September – October 2019) and 1-year follow-up (9–12 months after interventions had been delivered) (pre-COVID-19 group: January–11th March 2020, COVID-19 group: February–April 2021). Participants in phase 1 acted as controls. Those in phase 2 were exposed to the COVID-19 pandemic between the baseline and follow-up assessments, leading to a natural experiment.

## 2.2. Participants

Recruitment of participants was conducted in multiple stages. First, schools selected classes in relevant year groups to receive an intervention, if allocated. Second, letters were sent to the parents/carers of these pupils with information about the study; at this stage, they were offered the chance to opt their child out of the research. Finally, before completing the online surveys, pupils were presented with an information sheet and could assent to taking part by ticking all relevant boxes. If assent was not gained, the young person could not be part of the evaluation. The first young person participated on 17 September 2018. Ethics approval was obtained from University College London Research Ethics Committee [6735/009, 6735/014].

The main analytic sample in the current study was defined as all schools that were recruited to the trial and that took part in pupil surveys at both timepoints (baseline and 1-year follow up). All participants who completed some items of the survey at baseline or 1-year follow up were considered as part of the primary analysis sample. Figure 2 illustrates the participant flow diagram including drop out at various stages and the final analytic sample. A small number of unmatched schools were lost during linkage to the National Pupil Database (NPD). Similarly, pupils that could not be matched to the NPD were excluded and those who did not have data on the covariates of interest.

A total of 11 450 pupils from 178 schools were included in the main analytic sample with 6419 pupils from 90 schools in phase 1 (pre-COVID-19 group) and 5031 pupils from 88 schools in phase 2 (COVID-19 group). In phase 1, there was an average of 89.8 pupils per school (s.d. = 40.8; 10th percentile 44; 90th percentile 149; range 10–173); in phase 2, there were 78.9 pupils per school on average (s.d. = 41.0; 10th percentile 35; 90th percentile 139; range 3–185).

## 2.3. Variables

### 2.3.1. Individual-level covariates

We examined group differences and controlled for a range of individual pupil-level characteristics. These included age group (school year 7, 8 or 9 at baseline), child gender (male or female), socio-economic position assessed using eligibility for free school meals (FSM eligible or not), ethnicity (white or ethnic minority) and special education needs status (SEN, yes or no). Year group and gender data were provided directly by the schools and all other information on pupil-level socio-demographic characteristics were available via linkage to the NPD (table 1).

### 2.3.2. School level covariates

When investigating the effect of condition (pre-COVID-19 group versus COVID-19 group) on adolescent mental health outcomes, we also controlled for several school-level characteristics that were used for minimization for randomization following baseline data collection. This information was obtained from the Department for Education's Get Information About Schools (GIAS) service. School-level FSM eligibility (%) was included as an indicator of deprivation, as well as urban/rural status. The extent of existing mental health provision (prior interventions) reported by schools at baseline was also included as a covariate. See table 1 for the source and coding for each school-level variable.

## 2.4. Measures

### 2.4.1. Outcome measures

At all timepoints and across both phases of the study, schools were instructed to administer the pupil questionnaires via a secure online survey in teacher-facilitated sessions during the normal school day. As described in the introduction, the primary outcome was depressive symptoms with externalizing difficulties and life satisfaction considered secondary outcomes. Reliability estimates (Omega ($\omega$)) were calculated for all outcomes at baseline for the study sample and can be found in electronic supplementary material, table S1.

To examine whether outcome measures were invariant across phases and that exposure to the COVID-19 pandemic did not result in adolescents interpreting and responding to measures differently, multigroup confirmatory factor analyses (CFA) were conducted using the WLSMV estimator due to ordinal item responses. First, single-factor CFAs were conducted for outcomes at baseline and follow-up for both phases to confirm their unidimensional structure. Configural and scalar invariance were

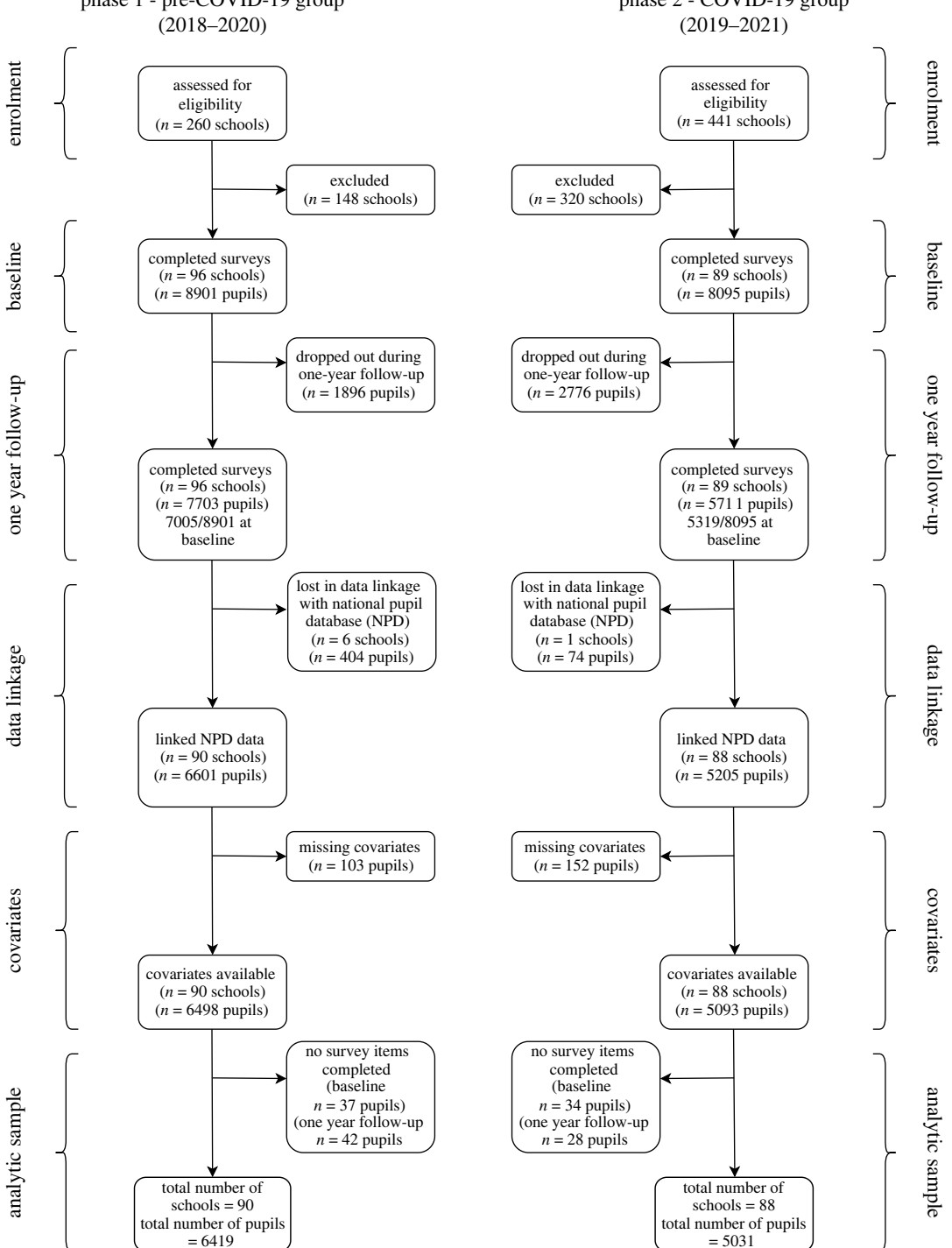

**Figure 2.** Flowchart of the current study—natural COVID-19 experiment.

then assessed by first freely estimating factor loadings and thresholds and then fixing them across phases [23]. Models were compared using the Mplus command DIFFTEST. For full results of the invariance testing see electronic supplementary material, tables S2a and S2b. School clustering was accounted for in all models using the type = COMPLEX command in Mplus.

### 2.4.2. Primary outcome: depressive symptoms

The primary outcome measure of this study was adolescent self-reported depressive symptoms. Participants completed the Short Mood and Feelings Questionnaire (SMFQ) [24], a 13-item self-report

**Table 1.** Covariate variables used in the analysis, data source, scoring and role in the analysis.

| variable | source | scoring | role |
|---|---|---|---|
| *school level* | | | |
| phase | based on phase of recruitment | 0 = phase 1 (pre-COVID-19 group) | focal variable |
| | | 1 = phase 2 (COVID-19 group) | |
| urban/rural | GIAS service, previously known as Edubase | 0 = Town/rural | covariate |
| | | 1 = City | |
| school deprivation − free school meal (FSM) eligibility | GIAS service, previously known as Edubase | % of pupils in school | covariate |
| prior interventions (baseline school mental health provision) | baseline school mental health provision survey | 0 = no mental health intervention/support | covariate |
| | | 1 = prior interventions structured lessons/other mental health support | |
| *individual level* | | | |
| FSM eligibility | NPD − code EVERFSM_ALL | 0 = not eligible | covariate (and modifier) |
| | | 1 = eligible | |
| gender | direct from schools | 0 = male | covariate (and modifier) |
| | | 1 = female | |
| special educational needs (SEN) | NPD − code SENprovisionMajor | 0 = No SEN | covariate (and modifier) |
| | | 1 = SEN | |
| ethnicity | NPD − code ethnicgroupmajor | 0 = white | covariate (and modifier) |
| | | 1 = non-white ethnic minority | |
| age (year group) | direct from schools | year 7, 8 or 9 | covariate |
| baseline mental health outcome score | from baseline survey | centred score | covariate |

*Note.* NPD = National Pupil Database, GIAS = Get Information About Schools service.

measure of depressive symptoms in the previous two weeks. Examples of included items are 'I felt miserable or unhappy' and 'I felt I was no good anymore'. Items are rated on a 3-point Likert scale (0 = 'not true', 1 = 'sometimes', 2 = 'true'). Possible total scores range from 0 to 26, with higher scores indicating greater depressive symptoms. The continuous depressive symptoms total score was the primary outcome measure. A binary score to assess impact on prevalence of caseness was also generated using the established cut-off score (greater than or equal to 12) indicating high levels of depressive symptoms [24].

### 2.4.3. Secondary outcomes: externalizing difficulties and life satisfaction

Externalizing difficulties were measured using the behavioural difficulties subscale of the Me and My Feelings questionnaire [25,26], a six-item self-report scale (e.g. 'I hit out when I'm angry') with three response options: 'never', 'sometimes', and 'always'. Responses were summed to create a total behavioural difficulties score, with higher scores indicating greater difficulties. The scale has an established cut-off score of greater than or equal to 6, which was used for analysis examining a binary outcome of high externalizing difficulties [25].

Life satisfaction was measured using the Huebner Life Satisfaction Scale (LSS; [27]). The original scale consisted of 10 items with four response options: 'never', 'sometimes', 'often' and 'almost always'. In this

study, we used the adapted version of the scale that was reduced to seven items with six-point Likert scales ranging from 'strongly disagree' to 'strongly agree' [27,28]. Items include 'My life is going well' and two items, 'I would like to change many things in my life' and 'I wish I had a different kind of life' that were reverse scored so that high scores indicated greater life satisfaction. A total score was created summing responses from the seven items, with higher total scores indicating greater life satisfaction.

## 2.5. Analysis strategy

### 2.5.1. Descriptive statistics and data checks

The distribution of baseline characteristics was compared for the schools across the two phases, using effect sizes to describe potentially relevant univariate differences. The same comparison was also performed for participants and individual pupil-level characteristics.

Descriptive statistics (means for continuous and percentages for binary outcomes) with 95% confidence intervals are presented for the primary and secondary outcomes at baseline and follow-up for each phase. We also present histograms showing the distributions of the continuous outcome variables at baseline and follow-up for each phase in the electronic supplementary material file.

### 2.5.2. Estimating the impact of the pandemic

All analyses were conducted using Stata 17 software. The primary outcome analysis used a random intercept (for schools) linear multivariable regression model with depressive symptoms at 1-year follow-up as the dependent variable. The model was specified as follows:

$$\text{Level 1:} \quad DS_{ij} = \beta_{0j} + \beta_{(1-k)}\text{Indiv}IV_{ij} + e_{ij}$$

$$\text{Level 2:} \quad \beta_{0j} = \gamma_{00} + \beta_{01}\text{Phase}_j + \beta_{0(2-l)}\text{School}IVs_j + u_{0j}$$

with $DS_{ij}$ as the depressive symptom score at 1-year follow-up of student $i$ in school $j$. On Level 1, the depression score was regressed upon fixed effects of the $k$ individual student covariates (*IndivIV*; table 1); and on Level 2 on $l$ school characteristics (table 1) with school-level variance $u_{0j}$ and individual-level error term $e_{ij}$. As all Level-1 slopes were defined as fixed effects, the additional specification equations were omitted. *Phase* (exposure to the COVID-19 pandemic) was the key regressor and the primary result was the coefficient $\beta_{01}$, which, if found statistically significant at $p < 0.05$, can be interpreted as the COVID-19 pandemic having potentially had an effect on adolescents' depressive symptoms. The direction of the potential effect is 'increased adolescent depressive symptoms' if the SMFQ score is higher in phase 2 (and 'decreased' if higher in phase 1). If the coefficient is not statistically significant, the primary outcome analysis can be interpreted as 'no supporting evidence for a difference was found'. For the dichotomous high depressive symptoms and externalizing symptoms, similar random intercept logistic multivariable regression models were conducted, and odds ratios reported.

The same strategy was employed for the secondary outcomes, but the primary outcome analysis takes priority in interpretation of the results. As the selection of independent variables was determined by the variables available in the main study, the reporting focuses on the result for the focal variable - phase. While the full regression models are reported as supplementary information, the coefficients for the other variables are not interpreted [29].

Standardized effect sizes were estimated for all three outcomes (one primary and two secondary outcomes) by dividing the estimated coefficient $\boldsymbol{\beta}_{01}$ over the standard deviation of the dependent variable. Apart from providing estimates of practical significance of the findings, this has the additional benefit of allowing comparisons of effect sizes across the outcomes (which is not possible when comparing coefficients as the scales and ranges of the measures vary). While it is difficult to provide a general cut-off for what is a relevant effect size in population-based research, in this study anything above 10% of a standard deviation change in continuous scores can be considered an effect with potentially practical significance at population level [30].

An additional approach to considering effect size as recommended for population-based research [30], is the population attributable fraction (PAF; [31]). We estimated the PAF only for the primary outcome and given there was support for our hypothesis (i.e. impact of COVID-19 on depressive symptoms). This allows one to estimate the number of cases that are attributable to the exposure of

interest (i.e. the COVID-19 pandemic) and hence from this we estimate the proportion of fewer cases that might be expected in the absence of the COVID-19 pandemic. This was estimated using the punaf package for Stata [32].

### 2.5.3. Sensitivity analyses

To describe the multivariate comparability of the samples across the two phases considering school and individual-level variables, a propensity score was estimated, first for individual level only and then for a model with both school and individual-level baseline variables as predictors of phase. We separately visualized the distribution of the two propensity scores across the cohorts using the Stata module psmatch2 (pstest; [33]) and reported descriptive statistics for the included covariates (table 1), and the estimated bias in the covariates between phases, with and without matching. This descriptive method offers comprehensive insight into the comparability of the underlying samples with respect to the available characteristics.

While the applied mixed-effects models can accommodate missing data in the dependent variable (under the MAR assumption), two sensitivity analyses explored the sensitivity of our main finding by adding additional approaches to account for missing data. The first of these analyses extended all models by inverse probability weighting for the probability to be a drop-out at follow-up (see below the description of missing data analyses). The second of these analyses used multiple imputation with chained equations with the full set of study variables as auxiliary variables to enhance the data for participants who were only partially observed at baseline. Results of the primary and secondary outcomes, including the two sensitivity analyses, are reported.

### 2.5.4. Exploratory analysis: subgroup differences

To examine whether the impacts of the pandemic were differently experienced by sub-groups of adolescents, the main analysis was conducted with an interaction term between each modifier of interest and phase in a separate model for each modifier (gender, ethnicity, socio-economic status and SEN). Given the modifiers of interest are all binary categorical (coded 0,1) these were entered into the models as is, as these interaction terms remain directly interpretable. If an interaction term had a $p$-value < 0.10 we visualized the interaction results in a graph based on predicted margins from the model. A sensitivity analysis for these models was also conducted where all modifiers and their interaction with phase were included in the same model [34].

### 2.5.5. Missing data

We first examined the rates and predictors of missing data at follow-up assessment. Alongside the complete case analysis, we (1) conducted weighted analysis to account for non-response at follow-up. Non-response weights were created separately in each phase as the mechanisms generating missingness might have varied between phases; and (2) used multiple imputation with chained equations with clustering due to schools to impute data for participants who were partially observed at the baseline assessments (i.e. took part in the survey but did not complete the primary or secondary outcome measures).

### 2.5.6. Power analysis

The prospective evaluation of statistical power was constrained by the design of the original studies for which power analyses were published [21,22]. As in the main project, we assumed for the primary outcome, self-reported depressive symptoms as measured by the SMFQ, a conservative school-level intraclass correlation of $\rho = 0.10$. Based on our current estimate of the database we expected around 185 analysed schools (phase 1, $n = 96$; phase 2, $n = 89$), and an average of 72.5 students per school (figure 2) and accepting a significance level of $p = 0.05$ and statistical power of $\beta = 0.80$, the minimally detectable effect size (MDES; [35]) is estimated as MDES = 0.139 (all estimates obtained with Optimal Design; [35]). Assuming potential additional losses on student-level of 10% due to inability to match data with NPD records increases this to MDES = 0.140.

For dichotomized SMFQ values, the analysis evaluates whether the share of students with a changed score differs between the two phases at follow-up. Based on estimates obtained with the same measure in the population-based Millennium Cohort Study [36], we assumed a plausible range for the prevalence of increased levels of depressive symptoms before the pandemic was between 0.10 and 0.25. Expecting a

pre-pandemic prevalence point estimate of 0.15, the prevalence after the pandemic would need to be either lower than 0.128 or above 0.174 to be detectable accepting a significance level of $p = 0.05$ and statistical power of $\beta = 0.80$ (0.127 and 0.175 for 10% of student-level dropout). The addition of covariates with predictive power on any level (school or pupil), potentially further increased precision of estimates. The primary outcome analysis was well-powered to identify a potentially relevant effect of the COVID-19 pandemic and the societal response on pupils' depressive symptoms.

### 2.5.7. Deviations from the registered report and unplanned exploratory analysis

To describe the multivariate comparability of the samples across the two phases considering school and individual-level variables, we had originally intended to use a random intercept logistic regression model. However, since schools were nested within phases, there was no variance within level-2 units (schools) and this was not statistically possible. We, therefore, did not account for the multi-level design when estimating propensity scores, but we did generate a propensity score for individual-level characteristics only and then with both school and individual-level baseline variables predicting phase as originally planned.

Following all planned data analyses, we conducted non-preregistered psychometric analyses with all outcome measures to examine whether exposure to the COVID-19 pandemic resulted in adolescents interpreting and responding to measures differently. Measurement invariance was assessed using a structural equation modelling (SEM) framework.

An additional exploratory analysis (not included in Stage 1 registration) was planned prior to data access which focussed on prior mental health as a modifier (based on binary scores at baseline). This was conducted to examine whether the impacts of the pandemic were different for adolescents with pre-existing mental health difficulties.

# 3. Results

## 3.1. Descriptive statistics and data checks

### 3.1.1. Analysis of sample bias and differences in non-response by phase

Table 2 shows the descriptive statistics for the schools in the final analytic sample. There were approximately the same number of schools in both phases of the study and the distribution of these schools based on school-level characteristics was similar. Overall, individual-level characteristics for the final analytic sample were similar across phases (table 3). The sample representativeness was also assessed by comparing individual-level characteristics to the secondary school population in England (2020/21) [37]. Overall, there was a slightly higher proportion of pupils eligible for FSM (23.5% versus 18.9%), and approximately the same proportion of pupils with special educational needs (10.4% versus 11.5%). Ethnic minority pupils were slightly underrepresented (25% compared to 32.1%).

Electronic supplementary material, table S3 describes the individual-level characteristics of participants at baseline and 1-year follow-up in both phases prior to the analytic sample selection. Propensity scores were used to examine differences between the two phases (pre-COVID-19 group versus COVID-19 group) on covariate characteristics. The balance between the two phases was estimated using two propensity score models (1) based only on the individual-level covariates and (2) based on both the individual and the non-regional school-level covariates. Electronic supplementary material, table S4 shows the results of the models and indicates no bias in the matched sample when only including the individual-level covariates. When matching on both the individual and school-level covariates, some differences are observed in the matched samples across phases by FSM eligibility, school-level FSM middle tertile, and year group 9 which are over-represented in phase 2 of the study (COVID-19 group). However, the extent of the bias is small, with all bias estimates under 10%. Supplementary figures S1 and S2 show the distribution of the propensity score when created using individual-level covariates (electronic supplementary material, figure S1) and both individual and school-level covariates (electronic supplementary material, figure S2).

Rates and predictors of missing data at follow-up assessment were assessed by phase. There was greater pupil drop-out at follow-up in the COVID-19 group (pre-COVID-19 group $N = 1896$, COVID-19 group $N = 2776$). However, the following characteristics predicted non-response in both groups: pupils' FSM eligibility, SEN status, baseline externalizing difficulties, school-level FSM eligibility and urban school location. Pupils' baseline externalizing difficulties and being from a control (usual

**Table 2.** Descriptive statistics (count, %) of school-level characteristics for each phase in the analytic sample ($N = 178$ schools).

| school characteristics | school count (%) | | effect size Phi-coefficient ($\phi$) |
| | phase 1 | phase 2 | |
| | pre-COVID-19 group (control group) | COVID-19 group | |
| no. of schools | 90 | 88 | |
| *free school meal (FSM) eligibility* | | | |
| bottom third | 29 (32.2%) | 26 (29.6%) | 0.04 |
| middle third | 30 (33.3%) | 32 (36.4%) | |
| upper third | 31 (34.4%) | 30 (34.1%) | |
| *geographical location* | | | |
| London and surrounding area | 35 (38.9%) | 38 (43.2%) | 0.22 |
| Greater Manchester and the Northwest | 20 (22.2%) | 9 (10.2%) | |
| Bath and Bristol and surrounding area | 17 (18.9%) | 11 (12.5%) | |
| Durham and surrounding area | 18 (20.0%) | 30 (34.1%) | |
| *school location* | | | |
| city | 35 (38.9%) | 51 (58.0%) | 0.19 |
| town/rural | 55 (61.1%) | 37 (42.0%) | |
| *prior interventions* | | | |
| structured lessons/other mental health support | 45 (50.0%) | 48 (45.5%) | 0.05 |
| no mental health intervention/support | 45 (50.0%) | 40 (54.6%) | |
| *condition in trial* | | | |
| control (usual provision) | 31 (34.4%) | 31 (35.2%) | 0.01 |
| intervention | 59 (65.6%) | 57 (64.8%) | |

provision) school in the main trial predicted drop-out in the control group only. Whereas non-response in the COVID-19 group was predicted by being female and year group. For results of the logistic regressions predicting non-response by phase, see electronic supplementary material, table S5.

### 3.1.2. Descriptive statistics for the outcomes at baseline and follow-up for each phase

Table 4 shows the descriptive statistics for the primary and secondary outcomes at baseline and 1-year follow-up for each phase. Baseline depressive symptoms were higher in phase 2 compared with phase 1 and in both phases, depressive symptoms increased from baseline to 1-year follow-up. Levels of externalizing difficulties were similar at baseline and showed a slight increase at 1-year follow-up in both phases. Baseline life satisfaction scores were also similar across phases, with reduced scores between baseline and 1-year follow-up in both groups. Electronic supplementary material, figure S3 shows the distribution of outcome scores using histograms at baseline and 1-year follow-up across the two phases.

## 3.2. Estimating the impact of the pandemic

The main analyses used random intercept (for schools) linear multivariable regression models with outcomes at 1-year follow-up as the dependent variables. Using multi-level models accounted for pupils nested within schools. For all models, the main exposure was phase with individual and school-level characteristics, as well as baseline mental health, included as control variables. Table 5 shows the main results for phase (i.e. the impact of exposure to the COVID-19 pandemic) from models predicting the 1-year follow-up primary and secondary mental health outcomes while controlling for centred baseline mental health scores and the full set of individual and school-level

**Table 3.** Descriptive statistics (count, %) of individual-level characteristics for each phase in the analytic sample ($N = 11\,450$ pupils).

| individual characteristics | individual count (%) | | effect size phi-coefficient ($\phi$) |
|---|---|---|---|
| | phase 1 | phase 2 | |
| | pre-COVID-19 group (control group) | COVID-19 group | |
| no. of pupils | 6419 | 5031 | |
| *gender* | | | |
| male | 2961 (46.1%) | 2244 (44.6%) | 0.02 |
| female | 3458 (53.9%) | 2787 (55.4%) | |
| *age (year group at baseline)* | | | |
| year 7 | 1772 (27.6%) | 984 (19.6%) | 0.11 |
| year 8 | 1382 (21.5%) | 963 (19.1%) | |
| year 9 | 3265 (50.9%) | 3084 (61.3%) | |
| *ethnicity binary* | | | |
| white | 4990 (77.7%) | 3594 (71.4%) | 0.07 |
| ethnic minority | 1429 (22.3%) | 1437 (28.6%) | |
| *ethnic minority subgroups* | | | |
| Asian | 720 (11.2%) | 752 (15.0%) | 0.08 |
| Black | 238 (3.7%) | 297 (5.9%) | |
| Mixed | 346 (5.4%) | 263 (5.2%) | |
| Chinese | 40 (0.6%) | 32 (0.6%) | |
| other | 85 (1.3%) | 93 (1.9%) | |
| *free school meal (FSM) eligibility* | | | |
| not eligible | 4908 (76.5%) | 3851 (76.6%) | 0.00 |
| eligible | 1511 (23.5%) | 1180 (23.5%) | |
| *special educational needs (SEN)* | | | |
| no SEN | 5736 (89.4%) | 4521 (89.9%) | 0.01 |
| SEN | 683 (10.6%) | 510 (10.1%) | |

covariates in the study. The full models, including coefficients for the full set of covariates, are included in the Supplementary File (electronic supplementary material, tables S6a–S6e).

There was a statistically significant effect of phase (exposure to the COVID-19 pandemic) on depressive symptoms and life satisfaction such that depressive symptoms were higher and life satisfaction scores lower in the group exposed to the COVID-19 pandemic (phase 2) when compared to the pre-COVID-19 control group (phase 1) (figure 3). For the continuous outcomes we also present coefficients based on standardized z scores to allow for comparable effect sizes across the outcomes. These coefficients suggest similar effect sizes for depressive symptoms and life satisfaction. Results from the two sensitivity analyses, including inverse probability weights for drop-out and multiple imputation for observed missingness at baseline produced very similar results to the main models (table 5).

Standardized effect sizes were estimated by dividing the coefficient for each outcome by the SD (akin to estimating d). Based on this, the effect sizes were 0.14 for depressive symptoms, less than 0.01 for externalizing difficulties, and −0.14 for life satisfaction.

### 3.2.1. Population attributable fraction

For the primary binary outcome, high depressive symptoms, the population attributable fraction (PAF) was calculated based on the scenario set at phase 1 (pre-COVID-19). Setting the baseline scenario as

**Table 4.** Descriptive statistics (means (standard deviations (SD)) for continuous and percentages (counts) for binary outcomes) with 95% confidence intervals (95% CI) for the primary and secondary outcomes at baseline and 1-year follow-up for each phase.

| | phase 1 pre-COVID-19 group (control group) | | phase 2 COVID-19 group | |
| --- | --- | --- | --- | --- |
| outcomes | baseline | follow-up | baseline | follow-up |
| *Depressive symptoms (primary outcome); n = 11 409* | | | | |
| mean (s.d.) | 5.93 (5.60) | 7.19 (6.65) | 6.53 (5.91) | 8.35 (6.74) |
| 95% CI | [5.8, 6.1] | [7.0, 7.4] | [6.4, 6.7] | [8.2, 8.5] |
| *high depressive symptoms (binary)* | | | | |
| % (count) | 16.4 (1050) | 24.5 (1569) | 19.7 (987) | 30.4 (1523) |
| 95% CI | [15.5, 17.3] | [23.5, 25.6] | [18.6, 20.8] | [29.1, 31.7] |
| *externalizing difficulties (secondary outcome)* | | | | |
| mean (s.d.) | 3.21 (2.35) | 3.30 (2.47) | 3.29 (2.37) | 3.35 (2.45) |
| 95% CI | [3.2, 3.3] | [3.2, 3.4] | [3.2, 3.4] | [3.3, 3.4] |
| *high externalizing difficulties (binary)* | | | | |
| % (count) | 15.1 (963) | 17.3 (1105) | 15.2 (761) | 17.2 (863) |
| 95% CI | [14.2, 16.0] | [16.4, 18.2] | [14.2, 16.2] | [16.2, 18.3] |
| *life satisfaction (secondary outcome)* | | | | |
| mean (s.d.) | 32.15 (6.98) | 30.87 (7.38) | 31.53 (7.23) | 29.59 (7.32) |
| 95% CI | [32.0, 32.3] | [30.7, 31.1] | [31.3, 31.7] | [29.4, 29.8] |

exposure to the COVID-19 pandemic (phase 2) and scenario 1 as phase 1 (pre-COVID-19) enables the estimation of the proportion of cases fewer than might be expected in the absence of the COVID-19 pandemic. The mean ratio of the PAF (i.e. those cases that can be attributed to the exposure to the COVID-19 pandemic) was 0.06. We, therefore, estimate that if the COVID-19 pandemic had not occurred, we would observe 6% fewer adolescents with high depressive symptoms which is a difference of 1.6% in prevalence. Given that the prevalence of high depressive symptoms in the data was 27.1%, it can be estimated that in a scenario where the COVID-19 pandemic did not happen, the prevalence would be 25.5%. The mean ratios and mean rate ratios are presented for the unattributable and attributable fraction in electronic supplementary material, table S7.

## 3.3. Subgroup differences

Effect modification was examined for the main findings by gender, FSM eligibility, ethnicity and SEN status. For each of the study outcomes, a separate model was run including an estimation of a phase by covariate interaction (e.g. phase × gender) (table 6).

For outcomes that indicated a possible interaction (as defined by having a *p*-value of < 0.10), figures were produced plotting the model predicted marginal means (and probabilities for binary outcomes) (figure 3). There was some evidence for effect modification. For example, a phase-by-gender interaction was observed for all continuous outcomes, such that females in phase 2 (COVID-19 group) had worse outcomes than in phase 1 (pre-COVID-19 group) with a greater difference between the control and COVID-19 group for females than males. Given that there was no significant main effect of phase on externalizing difficulties, the phase by gender interaction suggests that exposure to the COVID-19 pandemic only had a negative impact on girls' externalizing difficulties. There was also a phase by FSM eligibility interaction for life satisfaction. Both the control group and the COVID-19 group showed lower life satisfaction scores for adolescents eligible for FSM. However, adolescents of higher socio-economic position (not eligible for FSM) revealed a greater difference in life satisfaction between the control and COVID-19 group, with scores decreasing towards levels reported by the FSM eligible group. We also visualized marginal mean interactions for phase by ethnicity predicting

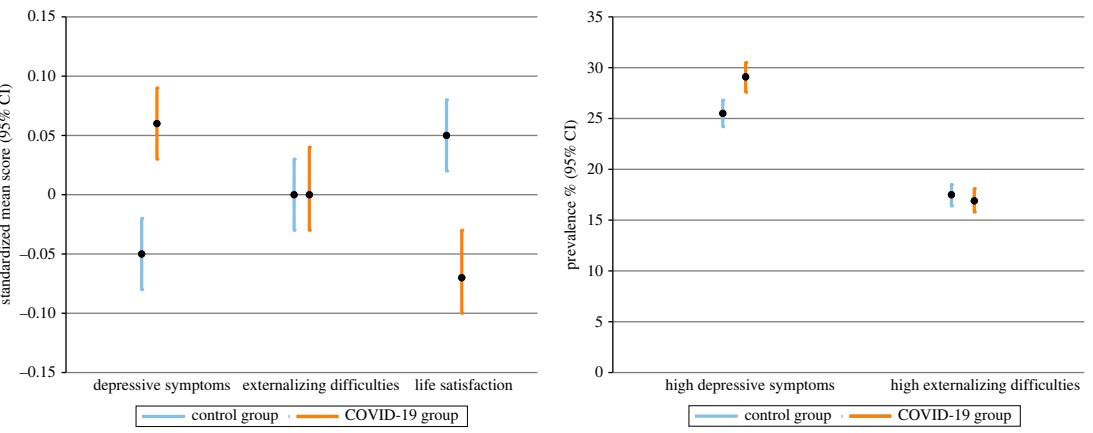

Note. Means are model predicted (controlling for baseline scores and other covariates) and standardised to enable comparison across mental health outcomes. Prevalence rates are calculated by multiplying model predicted probabilities by 100.

**Figure 3.** Difference between the pre-COVID-19 group and the COVID-19 group at 1-year follow-up for all mental health outcomes.

externalizing symptoms and phase by SEN status predicting high depressive symptoms. As can be seen in figure 4, no clear interactions can be observed in these visualizations.

As a sensitivity check of the effect modification analyses, we also estimate a model including all covariate interaction terms with phase in the same model (see electronic supplementary material, table S8; [34]). Some of the interactions observed in table 6 are attenuated when included jointly in a model with other interaction terms (e.g. phase by gender predicting depressive symptoms).

### 3.4. Non-preregistered analysis

Results from the additional exploratory analyses examining whether the impacts of the pandemic were different for adolescents with pre-existing mental health difficulties, do not suggest that the main effect for phase is modified by prior mental health.

Measurement invariance by phase was examined and full results of the invariance testing can be found in electronic supplementary material, tables S2a and S2b.

## 4. Discussion

This paper aimed to provide robust evidence on the impact of the COVID-19 pandemic on adolescent mental health, specifically, depressive symptoms (primary outcome), externalizing difficulties and life satisfaction (secondary outcomes). We also aimed to investigate, in exploratory analyses, whether there were socio-demographic differences (based on gender, ethnicity, socio-economic disadvantage and special educational needs) in the impact of the COVID-19 pandemic on adolescent mental health outcomes. Results revealed that the COVID-19 pandemic has led to an increase in adolescent depressive symptoms and a decrease in life satisfaction. After controlling for baseline scores and several school and pupil-level characteristics, depressive symptoms were higher and life satisfaction scores lower in the group exposed to the COVID-19 pandemic. We estimate that if the COVID-19 pandemic had not occurred, we would observe 6% fewer adolescents with high depressive symptoms. The pandemic has therefore led to a deterioration of mental health in this population beyond what would have been expected based on existing trends. However, there was no main effect of the COVID-19 pandemic on adolescent externalizing difficulties. Exploratory analyses suggest that the impact of the pandemic may have been greater in females, with females exposed to the pandemic showing greater depressive symptoms, externalizing difficulties and lower wellbeing. Adolescents of higher socio-economic position showed a greater difference in life satisfaction between the control and COVID-19 group.

The current study analysed mental health and wellbeing at baseline and 1-year follow-up for two comparable groups, recruited into the study in two phases. The inclusion of two groups with multiple timepoints provides an indication of the increase in adolescent depressive symptoms over a 1-year period. Adolescents recruited in phase 1 were not exposed to the COVID-19 pandemic between their baseline and 1-year follow-up mental health assessments. This group, therefore, acted as controls. By

**Table 5.** Results for the main exposure (phase) including coefficients for continuous outcomes (coef) and odds ratios (OR) for binary outcomes and their 95% confidence intervals (95% CI).

| all coefficients presented are for Phase (exposure to the Covid-19 pandemic compared with the control group) | depressive symptoms coef [95% CI] n, p-value | high depressive symptoms (binary) OR [95% CI] n, p-value | externalizing difficulties coef [95% CI] n, p-value | high externalizing difficulties (binary) OR [95% CI] n, p-value | life satisfaction coef [95% CI] n, p-value |
|---|---|---|---|---|---|
| *covariate-adjusted main models* | 0.77 [0.5, 1.1] 11 368, <0.001 | 1.26 [1.1, 1.4] 11 368, <0.001 | 0.01 [−0.1, 0.1] 11 353, 0.816 | 0.95 [0.8, 1.1] 11 353, 0.484 | −0.87 [−1.2, 0.6] 11 103, <0.001 |
| *standardised continuous outcomes for comparable effect sizes (adjusted)* | 0.11 [0.1, 0.2] 11 368, <0.001 | — | 0.01 [−0.0, 0.1] 11 353, 0.816 | — | −0.12 [−0.2, −0.1] 11 103, <0.001 |
| *covariate-adjusted + IPW for drop-out* | 0.78 [0.5, 1.1] 11 222, <0.001 | 1.26 [1.1, 1.4] 11 222, <0.001 | −0.00 [−0.1, 0.1] 11 220, 0.937 | 0.93 [0.8, 1.1] 11 220, 0.297 | −0.83 [−1.2, −0.5] 11 045, <0.001 |
| *covariate-adjusted + MI for observed missing at baseline* | 0.77 [0.5, 1.1] 11 474, <0.001 | 1.26 [1.1, 1.4] 11 474, <0.001 | 0.01 [−0.1, 0.1] 11 472, 0.868 | 0.95 [0.8, 1.1] 11 472, 0.423 | −0.86 [−1.2, −0.6] 11 279, <0.001 |

*Note.* All models adjusted for school-level and individual-level variables listed in table 1; IPW = inverse probability weight; MI = multiple imputation, fully conditional.

**Table 6.** Effect modification: interaction analyses by individual gender, ethnicity, FSM eligibility, and SEN including coefficients (coef; for continuous outcomes) and odds ratios (OR; for binary outcomes) and their 95% confidence intervals (95% CI).

| interaction | depressive symptoms coef [95% CI] p-value | high depressive symptoms (binary) OR [95% CI] p-value | externalizing difficulties coef [95% CI] p-value | high externalizing difficulties (binary) OR [95% CI] p-value | life satisfaction coef [95% CI] p-value |
|---|---|---|---|---|---|
| phase × gender | 0.36 [−0.1, 0.8] | 1.06 [0.9, 1.3] | 0.36 [0.2, 0.5] | 1.46 [1.2, 1.8] | −0.48 [−1.0, 0.0] |
| (female) | 0.009 | 0.590 | <0.001 | <0.001 | 0.051 |
| phase × FSM | −0.09 [−0.6, 0.4] | 0.96 [0.8, 1.2] | −0.03 [−0.2, 0.2] | 1.00 [0.8, 1.3] | 0.51 [−0.1, 1.1] |
| eligibility (eligible) | 0.735 | 0.750 | 0.792 | 0.973 | 0.078 |
| phase × ethnicity | −0.04 [−0.6, 0.4] | 0.88 [0.7, 1.1] | −0.20 [−0.4, −0.0] | 0.71 [0.6, 0.9] | −0.20 [−0.8, 0.4] |
| (ethnic minority) | 0.888 | 0.297 | 0.047 | 0.013 | 0.501 |
| phase × SEN | −0.41 [−1.1, 0.3] | 0.77 [0.6, 1.1] | −0.13 [−0.4, 0.1] | 0.94 [0.7, 1.3] | 0.26 [−0.5, 1.0] |
| (SEN = yes) | 0.240 | 0.096 | 0.307 | 0.738 | 0.510 |
| *additional interaction analysis with prior mental health* | | | | | |
| phase × prior MH | −0.11 [−0.7, 0.4] | 0.92 [0.7, 1.2] | 0.03 [−0.2, 0.3] | 1.05 [0.8, 1.3] | 0.33 [−0.3, 0.9] |
| (Prior MH = high) | 0.680 | 0.490 | 0.783 | 0.716 | 0.294 |
| N for models | 11 368 | 11 368 | 11 353 | 11 353 | 11 103 |

Note. The coefficient reported is for the interaction term between phase and each modifier and the p-value. The results from any model indicating the presence of effect modification (p < 0.1) will be described in a visualization. All models adjusted for school-level and individual-level variables listed in table 1; IPW = inverse probability weight; MI = multiple imputation, fully conditional; FSM = free school meal; SEN = special educational needs; MH = mental health.

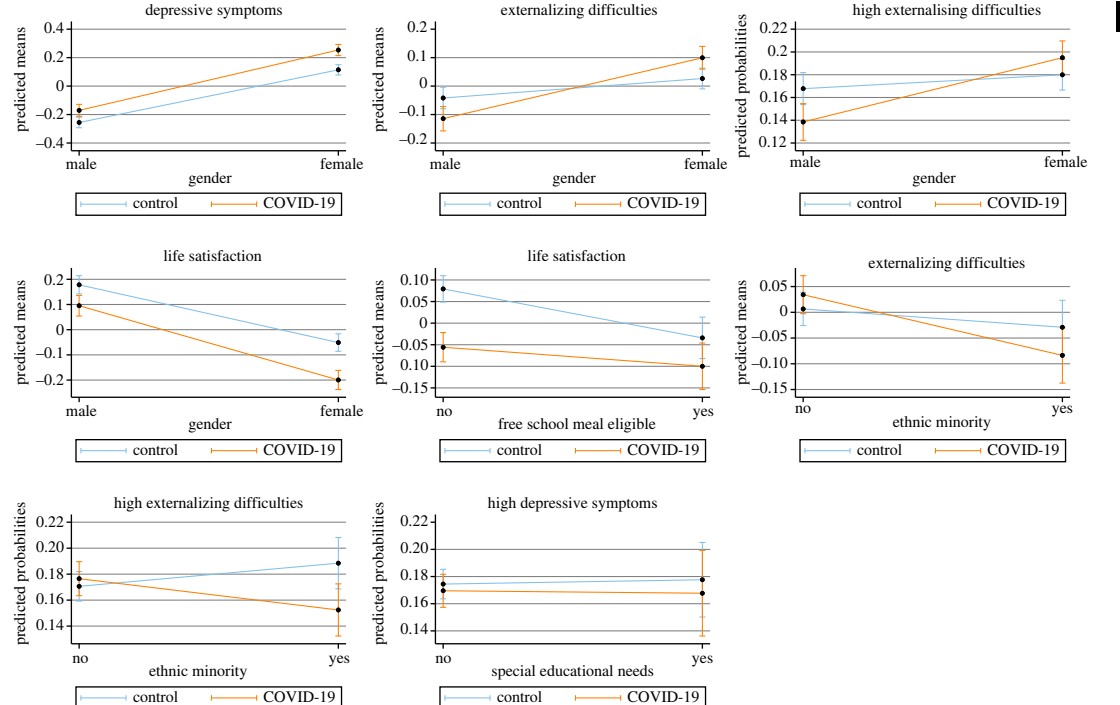

Note. The results are based on the standardised outcome scores to allow comparisons of effect sizes. The main analysis was conducted with an interaction term between each modifier of interest and phase in a separate model for each modifier (gender, ethnicity, free school meal eligibility and special education needs), controlling for the baseline outcome score and all other school and individual-level covariates.

**Figure 4.** Model predicted marginal mean results (and probabilities for binary outcomes) illustrating potential interactions suggesting effect modification.

including a comparable control condition, this study was better able to isolate the impact of the pandemic from known developmental trends in adolescent depressive symptoms [10,11]. An increase in depressive symptoms from baseline to 1-year follow-up was observed in both the control group and the group exposed to the COVID-19 pandemic. However, after controlling for baseline scores and several school and pupil-level characteristics, depressive symptoms were higher in the COVID-19 group. These findings demonstrate that the COVID-19 pandemic increased adolescent depressive symptoms beyond what would have likely occurred under non-pandemic circumstances.

At the time of writing the registered report for this study, there was limited evidence in the UK based on longitudinal, adolescent data. International studies revealed mixed results, with fewer mental health difficulties reported in China [15] and the Netherlands [8]. However, results from the current study show similar impacts of the COVID-19 pandemic on adolescent mental health to those found in Australia and Iceland, where COVID-19 restrictions led to a significant increase in adolescent depressive symptoms and a significant decrease in wellbeing [13,16]. In Spain, there were increased adolescent conduct difficulties reported during the COVID-19 pandemic by parents [14]. No main effect of exposure to the pandemic was found for externalizing difficulties in the current study. This may potentially reflect a difference in self- versus parent-reported difficulties as opposed to unequal impacts in England compared to Spain.

The Wirral Child Health and Development Study, a regional UK birth cohort, also provided evidence to suggest that depressive symptoms and externalizing difficulties increased following the first COVID-19 lockdown [17]. However, the sample was slightly younger (aged 11–12 years) than the current study and disproportionately sampled higher risk families, with externalizing difficulties reported by mothers. Another study in the Southwest of England [38] found 13–14-year-old adolescents' mental health improved during COVID-19 restrictions and suggested this could be due to the removal of stressors in the school environment. Due to a lack of a comparable control condition, it is also difficult to unpick the impact of the pandemic from known developmental trajectories in these studies. At the national level, despite finding a higher proportion of children experiencing mental health difficulties during the pandemic, it is difficult to compare results from the COVID-19 follow-up of the 2017 prevalence survey to the current study due to the mode and method of mental health assessment, low response rate during the pandemic, and difficulties separating developmental changes from pandemic related impact [18,19].

In a recent review, Aknin and colleagues summarize the high-quality studies published during early 2021 on the mental health consequences of the COVID-19 pandemic [39]. Although neither specifically focused on the UK nor adolescent populations, the review concludes that psychological distress increased during the early stages of the pandemic. The results from the current study, therefore, support the wider body of literature based on repeated cross-sectional and longitudinal surveys.

Exploratory subgroup analyses revealed that the negative impact of the COVID-19 pandemic on adolescent mental health may have been greater for females than males. This finding is consistent with the existing literature, with female mental health consistently found to be worse than males during the pandemic [13,16,17,18]. Evidence from a UK study tracking families throughout the pandemic found that children with SEN and from low-income homes were particularly impacted by COVID-19 related school closures and lockdown [20]. Exploratory analyses in the current study did not find any significant subgroup effects for adolescents with SEN. Across both the control and the COVID-19 group, adolescents eligible for FSM showed lower life satisfaction than those not eligible. However, adolescents of higher socio-economic position (not eligible for FSM) showed a greater difference between life satisfaction scores in the control group and the COVID-19 group, with the eligible and not eligible adolescents that were exposed to the pandemic showing much closer levels of life satisfaction. Similar findings were reported by the Wirral Child Health and Development Study [17], with rates of depression increasing in the less deprived families to levels matching the high deprivation group. However, unlike their findings, in non-preregistered analyses by prior mental health we find no evidence in these data that adolescents with pre-existing mental health difficulties were more negatively impacted by the pandemic.

## 4.1. Strengths and limitations

The occurrence of a natural experiment and the analytic approach allow for an attribution of decreased adolescent mental health levels to the occurrence of the COVID-19 pandemic. The confidence with which the observed increases in depressive symptoms and decreases in life satisfaction can be attributed to the pandemic is further increased by finding the schools and participants involved in phase 1 (control group) and phase 2 (COVID-19 group) were well-balanced on a range of relevant characteristics known to predict mental health. The data were drawn from two, national-level RCTs; the current study, therefore, benefits from large samples of pupils attending schools across England.

Prior to analyses, we estimated an expected 185 schools in the analytic sample. The final figure was slightly lower with 173 schools across both phases. The smaller sample will have reduced our estimated statistical power and therefore the effect sizes would have needed to be larger for them to be detected. However, for externalizing difficulties, the only (secondary) outcome for which we did not detect a statistically significant effect, the effect size was very small and the reduction in sample size inconsequential for the reported findings. Although the two phases were well-balanced on several school and pupil-level characteristics, there was greater drop-out at 1-year follow-up in the COVID-19 group (pre-COVID-19 group $n = 1896$ [23.3%], COVID-19 group $n = 2776$ [34.3%]). Greater drop-out could have resulted in a biased sample at follow-up, with schools and individuals with certain characteristics more likely to participate. However, predictors of non-response across the control and the COVID-19 group were mostly overlapping and controlling for the probability of drop-out in the analyses did not change the conclusions.

No strong imbalances were found in the distribution of control (usual provision) and intervention schools across the two phases. However, the effectiveness of interventions could have been compromised during the pandemic. Assessments of measurement invariance across all outcomes showed that for the SMFQ, exposure to the COVID-19 pandemic did not appear to influence the way in which adolescents interpreted and responded to the measure at follow-up. However, for the behavioural difficulties subscale of the Me and My Feelings questionnaire (MMFQ – BD) and the LSS, violations of measurement invariance indicate that the pandemic may have influenced the interpretation and response to these measures. More psychometric analyses are needed to fully understand the impact of these violations on analyses that compare those exposed to the pandemic with the control group.

## 4.2. Conclusion

There was widespread concern prior to the pandemic regarding high rates of adolescent mental health difficulties in England. An extended period of austerity has left schools and Child and Adolescent

Mental Health Services (CAMHS) with insufficient resources and capacity to support the increasing number of mental health difficulties experienced by adolescents. The current study provides evidence for increased adolescent depressive symptoms and decreased life satisfaction as a result of the COVID-19 pandemic. Schools and CAMHS services were forced to adapt to the challenging circumstances of the pandemic. If there had been clearer guidance and increased funding for schools, this may have enabled better systems for supporting pupils' mental health during this period. Given that the pandemic is ongoing and that these negative impacts are likely long lasting with other potential adverse lifelong consequences [40], the promotion of young people's mental health and increased access to services relies on a properly resourced, public health approach that builds capacity within and between sectors.

Ethics. Ethical approval was obtained from University College London Research Ethics Committee [6735/009, 6735/014]. Informed consent was obtained from all participants.

Data accessibility. As per our trial protocols [21,22], the raw data can only be shared after the trials are complete. An anonymized quantitative dataset was originally due to be made available in 2022 once the trial had finished; however, due to disruption caused by the COVID-19 pandemic, this timeline has been delayed. Analysis code is shared as electronic supplementary material [41] and on the Open Science Framework alongside the full stage 2 paper, and the data will be made available on this same OSF project location after the main trial results are published.

Authors' contributions. R.M.: formal analysis, methodology, visualization, writing—original draft, writing—review and editing; J.S.: data curation, formal analysis, methodology, project administration, visualization, writing—original draft, writing—review and editing; J.D.: conceptualization, funding acquisition, project administration, writing—review and editing; D.H.: conceptualization, project administration, writing—review and editing; T.V.: project administration, writing—review and editing; J.B.: conceptualization, funding acquisition, methodology, supervision, writing—original draft, writing—review and editing; P.P.: conceptualization, formal analysis, funding acquisition, methodology, supervision, visualization, writing—original draft, writing—review and editing.

All authors gave final approval for publication and agreed to be held accountable for the work performed therein.

Conflict of interest declaration. The views expressed are those of the authors and not necessarily those of the Department for Education, England or its arm's-length bodies or other government departments. The authors declare no conflict of interest and take sole responsibility for the content of this article.

Funding. This work (the Education for Wellbeing Programme) was supported by the Department for Education, England (grant no. EOR/SBU/2017/015).

Acknowledgements. We would like to thank all the colleagues and researchers involved in the Education for Wellbeing Programme from which these data were drawn. We are grateful to all the schools and students who are part of this large study for their participation.

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
