## [Peer Review File · Royal Society Open Science]

Review History

RSOS-211114.R0 (Original submission)

Review form: Reviewer 1

Do you have any ethical concerns with this paper?

No

Recommendation?

Accept in principle

Comments to the Author(s)

This study looks very promising! Please consider including a more detailed breakdown of student ethnicity if possible (although I recognize the statistical issues of analyzing by more than a binary difference), and comment on any differences you find. I look forward to your results and discussion. Best wishes in your work going forward.

Review form: Reviewer 2

Do you have any ethical concerns with this paper?

No

Recommendation?

Accept with minor revision

Comments to the Author(s)

- The scientific validity of the research question(s)
 - o The research question is scientifically valid.
- The logic, rationale, and plausibility of the proposed hypotheses
 - o The general hypothesis is clear, plausible, and rationale (that Covid-19 impacted mental health). I would like to see the authors detail more explicitly what they think they will see in their analyses.
- The soundness and feasibility of the methodology and analysis pipeline (including statistical power analysis where applicable)
 - o The analyses are largely sound and feasible, though I do have some recommendations for additions, and I request some clarifications as well.
- Whether the clarity and degree of methodological detail would be sufficient to replicate exactly the proposed experimental procedures and analysis pipeline
 - o The analysis pipeline is not yet detailed enough (see below).
- Whether the authors provide a sufficiently clear and detailed description of the methods to prevent undisclosed flexibility in the experimental procedures or analysis pipeline
 - o I request some further clarification below. I would have loved to see a full working analysis pipeline with simulated data, though I understand that is not always feasible.
- Whether the authors have considered sufficient outcome-neutral conditions (e.g. positive controls) for ensuring that the results obtained are able to test the stated hypotheses.
 - o There are no positive controls in the proposed analyses, though I do not think that that is a weakness in this case.

This registered report describes a study in which the impact of Covid-19 on adolescent mental health in the UK will be examined in a longitudinal sample. The study is particularly notable for having both a control sample and a sample who was assessed prior to and after the Covid-19 pandemic. Having previously reviewed several cross-sectional reports on this subject, I am quite enthusiastic about a preregistered longitudinal study with both control and covid-19 exposed participants. This study appears methodologically sound and looks like it will be very well-powered. I do have some requests for further details from the authors on their analysis plan and some suggestions for additional sensitivity tests.

1. Introduction. I would like to see the authors state their hypotheses clearly. What do they think the most likely outcome will be?
2. Mixed effect models. Please specify which variables will be used to define random intercepts. Presumably these will be student ID and school ID (though this is not specified anywhere). I think these should be nested (eg. (1 | School/Student), though I could see an argument for them being crossed instead. Which will it be?
3. Covariates. How will Geographical region be coded? Will it be a dummy-coded variable? Or will it also be a random intercept?
4. Interactions.
 - a. Will variables be centered prior to forming the interaction term? (presumably yes, but this should be explicitly stated).
 - b. Will the authors perform follow-up tests to aid in interpreting the interactions (e.g. Johnson-Neyman/Region of Significance, proportion of interaction, or cross-point)?

5. Sensitivity analyses. I would like to see additional sensitivity analyses for interaction analyses, in which covariate x main variable interaction terms are additionally included (to control for potentially spurious interactions, see <https://doi.org/10.1016/j.biopsycho.2013.09.006>). For example, if the main interaction analysis is:

Depression = Covid x Gender + Covid + Gender + Age + SES etc...

The sensitivity analysis would look like:

Depression = Covid x Gender + Covid + Gender + Age + SES + Covid x Age + Covid x SES + Gender x Age + Gender x SES etc...

6. Alpha & multiple test correction.

a. Alpha appears set to 0.05 for the main effect analyses (depression, life satisfaction, extraversion), but 0.10 for the interaction analyses. Why this difference? The authors state they will plot interactions that are $p < 0.10$. What interpretation will they ascribe to these interactions? That they are significant?

b. Additionally, given that the authors appear to planning on performing 15 tests in total (1 primary + 2 secondary + 4 modifiers x 3 main effects), I think it is worth raising the question of multiple test correction, which is not addressed anywhere. With $\alpha = .1$, at least one of the interaction tests will be "significant", though I am not convinced that it will be meaningful. Are the authors planning on correcting for multiple comparisons? I am less concerned about the primary and secondary outcomes, as the authors have clearly stated how they will interpret these, as well as which is "primary".

7. Effect sizes. In the power analyses, please specify what kind of effect the "MDES" is (e.g. standardized regression coefficient).

8. Limitations. This study will give some indication of whether Covid-19 causally impacted adolescent's mental health in the UK. Can the authors lay out the limitations of this study and reasons why we will not be able to fully ascribe causation if their results are significant?

Review form: Reviewer 3 (Argyris Stringaris)

Do you have any ethical concerns with this paper?

No

Recommendation?

Accept with minor revision

Comments to the Author(s)

This is Argyris Stringaris commenting on The impact of the COVID-19 pandemic on adolescent mental health: a natural experiment.

This is a very worthwhile and thoughtful use of a natural experiment--two samples that should be comparable in principle completed close to each other, but one before the pandemic, the other with this second wave of data collection, within the pandemic.

The lines of argument of the authors seem sound as does their overall approach.

I have the following comments.

It was hard for me to follow the details of the mixed effects model. I am not sure how the random effects are specified. To me there seem to be two: one random effect is the schools, the other is individuals over time. Do the authors take both into account? Also, it would be good to specify clearly and unambiguously the covariates of interest in advance. Ideally, the complete regression equation should be presented here. A maximal random effects structure specification should be preferred and the criteria for fit/convergence declared (see Barr et al 2013 J Mem Lang, Bell et al Qual and Quand 2019).

My other recommendation would be to specify some effect size of interest. The p-value is a less meaningful here, particularly given the expected sample size. I believe the authors should specify some minimal effect of interest (there is guidance for this for the MFQs (eg Liu et al 2019 JAACAP). How many points of a difference would seem important?

I love the authors' sensitivity analyses plans, all three of them are very appropriate. Again though, I would like to have some a priori notion of when the samples would be deemed comparable.

Decision letter (RSOS-211114.R0)

Dear Dr Patalay

On behalf of the Editors, I am pleased to inform you that your Manuscript RSOS-211114 entitled "The impact of the COVID-19 pandemic on adolescent mental health: a natural experiment" deemed suitable for in-principle acceptance in Royal Society Open Science subject to minor revision in accordance with the referee and editor suggestions. Please find their comments at the end of this email.

The reviewers and handling editors have recommended publication, but also suggest some minor revisions to your manuscript. Therefore, I invite you to respond to the comments and revise your manuscript.

Please you submit the revised version of your manuscript within 7 days (i.e. by the 27-Jul-2021). If you do not think you will be able to meet this date please let me know immediately.

Full author guidelines can be found here <https://royalsocietypublishing.org/rsos/registered-reports#ReviewerGuideRegRep>.

Kind regards
 Professor Chris Chambers
 Royal Society Open Science
 openscience@royalsociety.org

Associate Editor Comments to Author (Professor Chris Chambers):

Three reviewers have now evaluated the Stage 1 manuscript. All are positive about the submission while also noting a number of areas that will need careful attention to achieve in-principle acceptance (IPA), including clarity and specificity of the hypotheses, details of the sampling and analysis plans, and specification / justification of expected effect sizes. To help make the design as clear as possible, please include in the Method section of the revised manuscript a completed study design table, as set out in this template <https://osf.io/sbmx9/>

Provided the authors are able to address all points comprehensively in a revision and response, IPA should be forthcoming without requiring further in-depth review.

Reviewer comments to Author:

Reviewer: 1

Comments to the Author(s)

This study looks very promising! Please consider including a more detailed breakdown of student ethnicity if possible (although I recognize the statistical issues of analyzing by more than a binary difference), and comment on any differences you find. I look forward to your results and discussion. Best wishes in your work going forward.

Reviewer: 2

Comments to the Author(s)

- The scientific validity of the research question(s)
 - o The research question is scientifically valid.
- The logic, rationale, and plausibility of the proposed hypotheses
 - o The general hypothesis is clear, plausible, and rationale (that Covid-19 impacted mental health). I would like to see the authors detail more explicitly what they think they will see in their analyses.
- The soundness and feasibility of the methodology and analysis pipeline (including statistical power analysis where applicable)
 - o The analyses are largely sound and feasible, though I do have some recommendations for additions, and I request some clarifications as well.
- Whether the clarity and degree of methodological detail would be sufficient to replicate exactly the proposed experimental procedures and analysis pipeline
 - o The analysis pipeline is not yet detailed enough (see below).
- Whether the authors provide a sufficiently clear and detailed description of the methods to prevent undisclosed flexibility in the experimental procedures or analysis pipeline
 - o I request some further clarification below. I would have loved to see a full working analysis pipeline with simulated data, though I understand that is not always feasible.
- Whether the authors have considered sufficient outcome-neutral conditions (e.g. positive controls) for ensuring that the results obtained are able to test the stated hypotheses.
 - o There are no positive controls in the proposed analyses, though I do not think that that is a weakness in this case.

This registered report describes a study in which the impact of Covid-19 on adolescent mental health in the UK will be examined in a longitudinal sample. The study is particularly notable for having both a control sample and a sample who was assessed prior to and after the Covid-19 pandemic. Having previously reviewed several cross-sectional reports on this subject, I am quite enthusiastic about a preregistered longitudinal study with both control and covid-19 exposed

participants. This study appears methodologically sound and looks like it will be very well-powered. I do have some requests for further details from the authors on their analysis plan and some suggestions for additional sensitivity tests.

1. Introduction. I would like to see the authors state their hypotheses clearly. What do they think the most likely outcome will be?
2. Mixed effect models. Please specify which variables will be used to define random intercepts. Presumably these will be student ID and school ID (though this is not specified anywhere). I think these should be nested (eg. (1 | School/Student)), though I could see an argument for them being crossed instead. Which will it be?
3. Covariates. How will Geographical region be coded? Will it be a dummy-coded variable? Or will it also be a random intercept?
4. Interactions.
 - a. Will variables be centered prior to forming the interaction term? (presumably yes, but this should be explicitly stated).
 - b. Will the authors perform follow-up tests to aid in interpreting the interactions (e.g. Johnson-Neyman/Region of Significance, proportion of interaction, or cross-point)?
5. Sensitivity analyses. I would like to see additional sensitivity analyses for interaction analyses, in which covariate x main variable interaction terms are additionally included (to control for potentially spurious interactions, see <https://doi.org/10.1016/j.biopsycho.2013.09.006>). For example, if the main interaction analysis is:

$$\text{Depression} = \text{Covid} \times \text{Gender} + \text{Covid} + \text{Gender} + \text{Age} + \text{SES} \text{ etc...}$$
 The sensitivity analysis would look like:

$$\text{Depression} = \text{Covid} \times \text{Gender} + \text{Covid} + \text{Gender} + \text{Age} + \text{SES} + \text{Covid} \times \text{Age} + \text{Covid} \times \text{SES} + \text{Gender} \times \text{Age} + \text{Gender} \times \text{SES} \text{ etc...}$$
6. Alpha & multiple test correction.
 - a. Alpha appears set to 0.05 for the main effect analyses (depression, life satisfaction, extraversion), but 0.10 for the interaction analyses. Why this difference? The authors state they will plot interactions that are $p < 0.10$. What interpretation will they ascribe to these interactions? That they are significant?
 - b. Additionally, given that the authors appear to planning on performing 15 tests in total (1 primary+ 2 secondary + 4 modifiers x 3 main effects), I think it is worth raising the question of multiple test correction, which is not addressed anywhere. With $\alpha = .1$, at least one of the interaction tests will be “significant”, though I am not convinced that it will be meaningful. Are the authors planning on correcting for multiple comparisons? I am less concerned about the primary and secondary outcomes, as the authors have clearly stated how they will interpret these, as well as which is “primary”.
7. Effect sizes. In the power analyses, please specify what kind of effect the “MDES” is (e.g. standardized regression coefficient).
8. Limitations. This study will give some indication of whether Covid-19 causally impacted adolescent’s mental health in the UK. Can the authors lay out the limitations of this study and reasons why we will not be able to fully ascribe causation if their results are significant?

Reviewer: 3

Comments to the Author(s)

This is Argyris Stringaris commenting on The impact of the COVID-19 pandemic on adolescent mental health: a natural experiment.

This is a very worthwhile and thoughtful use of a natural experiment--two samples that should be comparable in principle completed close to each other, but one before the pandemic, the other with this second wave of data collection, within the pandemic.

The lines of argument of the authors seem sound as does their overall approach.

I have the following comments.

It was hard for me to follow the details of the mixed effects model. I am not sure how the random effects are specified. To me there seem to be too: one random effect is the schools, the other is individuals over time. Do the authors take both into account? Also, it would be good to specify clearly and unambiguously the covariates of interest in advance. Ideally, the complete regression equation should be presented here. A maximal random effects structure specification should be preferred and the criteria for fit/convergence declared (see Barr et al 2013 J Mem Lang, Bell et al Qual and Quand 2019).

My other recommendation would be to specify some effect size of interest. The p-value is a less meaningful here, particularly given the expected sample size. I believe the authors should specify some minimal effect of interest (there is guidance for this for the MFQs (eg Liu et al 2019 JAACAP). How many points of a difference would seem important?

I love the authors' sensitivity analyses plans, all three of them are very appropriate. Again though, I would like to have some a priori notion of when the samples would be deemed comparable.

Author's Response to Decision Letter for (RSOS-211114.R0)

See Appendix A.

Decision letter (RSOS-211114.R1)

Dear Dr Patalay

On behalf of the Editor, I am pleased to inform you that your Manuscript RSOS-211114.R1 entitled "The impact of the COVID-19 pandemic on adolescent mental health: a natural experiment" has been accepted in principle for publication in Royal Society Open Science.

Please read the following email carefully

Your accepted Stage 1 manuscript has been publicly registered at:
<https://doi.org/10.17605/OSF.IO/B25DH>

You may now progress to Stage 2 and complete the study as approved.

Following completion of your study, we invite you to resubmit your paper for peer review as a Stage 2 Registered Report. Please note that your manuscript can still be rejected for publication at Stage 2 if the Editors consider any of the following conditions to be met:

- The results were unable to test the authors' proposed hypotheses by failing to meet the approved outcome-neutral criteria.

- The authors altered the Introduction, rationale, or hypotheses, as approved in the Stage 1 submission.
- The authors failed to adhere closely to the registered study procedures. Please note that any deviations from the approved study procedures must be communicated to the editor immediately for approval, and prior to the completion of data collection. Failure to do so can result in revocation of in-principle acceptance and rejection at Stage 2 (see complete guidelines for further information).
- Any post-hoc (unregistered) analyses were either unjustified, insufficiently caveated, or overly dominant in shaping the authors' conclusions.
- The authors' conclusions were not justified given the data obtained.

Please be sure to include the following statement at the end of the Abstract in your Stage 2 manuscript: "Following in-principle acceptance, the approved Stage 1 version of this manuscript was preregistered on the OSF at <https://doi.org/10.17605/OSF.IO/B25DH>. This preregistration was performed prior to data analysis."

We encourage you to read the complete guidelines for authors concerning Stage 2 submissions at <https://royalsocietypublishing.org/rsos/registered-reports#ReviewerGuideRegRep>. Please especially note the requirements for data sharing, reporting the URL of the independently registered protocol, and that withdrawing your manuscript will result in publication of a Withdrawn Registration.

Once again, thank you for submitting your manuscript to Royal Society Open Science and we look forward to receiving your Stage 2 submission in due course. If you have any questions at all, please do not hesitate to get in touch.

on behalf of Professor Chris Chambers (Registered Reports Editor, Royal Society Open Science)
openscience@royalsociety.org

Author's Response to Decision Letter for (RSOS-211114.R1)

See Appendix B.

RSOS-211114.R2

Review form: Reviewer 2

Is the manuscript scientifically sound in its present form?

Yes

Are the interpretations and conclusions justified by the results?

Yes

Is the language acceptable?

Yes

Do you have any ethical concerns with this paper?

No

Have you any concerns about statistical analyses in this paper?

No

Recommendation?

Major revision

Comments to the Author(s)

1) Whether the data are able to test the authors' proposed hypotheses by passing the approved outcome-neutral criteria (such as absence of floor and ceiling effects or success of positive controls)

- No - data are not available, though it is stated that they are. I am recommending a 'major' revision for this reason, as data availability is a condition of publication at this journal.

2) Whether the Introduction, rationale and stated hypotheses are the same as the approved Stage 1 submission

- Yes

3) Whether the authors adhered precisely to the registered experimental procedures

- Yes

4) Where applicable, whether any unregistered exploratory statistical analyses are justified, methodologically sound, and informative

- Yes

5) Whether the authors' conclusions are justified given the data

- Conclusions regarding the moderating effect of sex (an exploratory analysis without multiple testing correction) need to be tempered, particularly in the abstract (which is what most people will read). For example:

Exploratory analyses ... *suggest* that the negative impact of ...

Females *may have been* impacted by the pandemic more than males across *most* outcomes

Decision letter (RSOS-211114.R2)

Dear Dr Patalay:

On behalf of the Editor, I am pleased to inform you that your Stage 2 Registered Report RSOS-211114.R2 entitled "The impact of the COVID-19 pandemic on adolescent mental health: a natural experiment" has been deemed suitable for publication in Royal Society Open Science subject to minor revision in accordance with the referee suggestions. Please find the referees' comments at the end of this email.

The reviewers and Subject Editor have recommended publication, but also suggest some minor revisions to your manuscript. We invite you to respond to the comments and revise your manuscript. Below the referees' and Editors' comments (where applicable) we provide additional requirements. Final acceptance of your manuscript is dependent on these requirements being met. We provide guidance below to help you prepare your revision.

Please submit your revised manuscript and required files (see below) no later than 7 days from today's (ie 16-Mar-2022) date. Note: the ScholarOne system will 'lock' if submission of the revision is attempted 7 or more days after the deadline. If you do not think you will be able to meet this deadline please contact the editorial office immediately.

on behalf of Professor Chris Chambers
(Registered Reports Editor, Royal Society Open Science)
openscience@royalsociety.org

Associate Editor Comments to Author (Professor Chris Chambers):

Associate Editor: 1

Comments to the Author:

One of the original Stage 1 reviewers was available to assess the Stage 2 manuscript, and I have decided that the reviewer's evaluation, combined with my own reading, is sufficient for us to continue with an interim editorial decision. As you will see, the reviewer broadly judges the Stage 2 criteria to be met, with the exception that conclusions based on exploratory analyses should be appropriately restrained. This is comment is line with the discussion concerning outcome reporting that we had at the pre-review stage, so please attend to it carefully.

Concerning the issue of data availability, this doesn't fall under Stage 2 criterion 1 (as implied by the reviewer's comment), but the reviewer is correct that this is a more general requirement of publishing in RSOS, unless special circumstances apply. The RSOS admin team will liaise with you separately to ensure these requirements are met.

Comments to Author:

Reviewer: 2

Comments to the Author(s)

1) Whether the data are able to test the authors' proposed hypotheses by passing the approved outcome-neutral criteria (such as absence of floor and ceiling effects or success of positive controls)

- No - data are not available, though it is stated that they are. I am recommending a 'major' revision for this reason, as data availability is a condition of publication at this journal.

2) Whether the Introduction, rationale and stated hypotheses are the same as the approved Stage 1 submission

- Yes

3) Whether the authors adhered precisely to the registered experimental procedures

- Yes

4) Where applicable, whether any unregistered exploratory statistical analyses are justified, methodologically sound, and informative

- Yes

5) Whether the authors' conclusions are justified given the data

- Conclusions regarding the moderating effect of sex (an exploratory analysis without multiple testing correction) need to be tempered, particularly in the abstract (which is what most people will read). For example:

Exploratory analyses ... *suggest* that the negative impact of ...

Females *may have been* impacted by the pandemic more than males across *most* outcomes

===PREPARING YOUR MANUSCRIPT===

one version should clearly identify all the changes that have been made (for instance, in coloured highlight, in bold text, or tracked changes);

While not essential, it will speed up the preparation of your manuscript proof if you format your references/bibliography in Vancouver style (please see

<https://royalsociety.org/journals/authors/author-guidelines/#formatting>). You should include DOIs for as many of the references as possible.

===PREPARING YOUR REVISION IN SCHOLARONE===

<https://royalsociety.org/journals/authors/author-guidelines/#data>. You should ensure that you cite the dataset in your reference list. If you have deposited data etc in the Dryad repository,

please only include the 'For publication' link at this stage. You should remove the 'For review' link.

-- If you are requesting an article processing charge waiver, you must select the relevant waiver option (if requesting a discretionary waiver, the form should have been uploaded, see 'File upload' above).

-- If you have uploaded any electronic supplementary (ESM) files, please ensure you follow the guidance at <https://royalsociety.org/journals/authors/author-guidelines/#supplementary-material> to include a suitable title and informative caption. An example of appropriate titling and captioning may be found at https://figshare.com/articles/Table_S2_from_Is_there_a_trade-off_between_peak_performance_and_performance_breadth_across_temperatures_for_aerobic_scope_in_teleost_fishes_/3843624.

Author's Response to Decision Letter for (RSOS-211114.R2)

See Appendix C.

Decision letter (RSOS-211114.R3)

Dear Dr Patalay:

It is a pleasure to accept your revised Stage 2 Registered Report entitled "The impact of the COVID-19 pandemic on adolescent mental health: a natural experiment" in its current form for publication in Royal Society Open Science.

COVID-19 rapid publication process:

We are taking steps to expedite the publication of research relevant to the pandemic. If you wish, you can opt to have your paper published as soon as it is ready, rather than waiting for it to be published the scheduled Wednesday.

This means your paper will not be included in the weekly media round-up which the Society sends to journalists ahead of publication. However, it will still appear in the COVID-19 Publishing Collection which journalists will be directed to each week (<https://royalsocietypublishing.org/topic/special-collections/novel-coronavirus-outbreak>).

If you wish to have your paper considered for immediate publication, or to discuss further, please notify openscience_proofs@royalsociety.org and press@royalsociety.org when you respond to this email.

Thank you for your fine contribution. On behalf of the Editors of Royal Society Open Science, we look forward to your continued contributions to the journal.

on behalf of Professor Chris Chambers (Subject Editor)
openscience@royalsociety.org

Appendix A

Manuscript: RSOS-211114 – responses to editor and reviewer comments

The impact of the COVID-19 pandemic on adolescent mental health: a natural experiment

Editor/Reviewer Comments	Author Response and Action
Associate Editor Comments (Professor Chris Chambers)	
Three reviewers have now evaluated the Stage 1 manuscript. All are positive about the submission while also noting a number of areas that will need careful attention to achieve in-principle acceptance (IPA), including clarity and specificity of the hypotheses, details of the sampling and analysis plans, and specification / justification of expected effect sizes. To help make the design as clear as possible, please include in the Method section of the revised manuscript a completed study design table, as set out in this template https://osf.io/sbmx9/	Please see the table using the suggested template now added (page 10, Table 2).
Reviewer 1	
Please consider including a more detailed breakdown of student ethnicity, if possible (although I recognize the statistical issues of analyzing by more than a binary difference), and comment on any differences you find.	Thank you for this comment. We acknowledge that a more detailed assessment of ethnic differences in the impact of the COVID-19 pandemic would be useful. However, as you note, including multiple dummy variables for different groups will considerably increase the number of predictors in the model, and several of these subgroups will have small Ns. Although it is possible with the NPD data to apply the ONS approach to harmonising and reporting ethnicity in broader groupings (e.g., White, Black, South Asian, Mixed, Other) we risk issues of identifiability with the potential for low numbers in some groups that may not exceed the required thresholds for the Office for National Statistics (ONS). Reporting of ethnicity data underlies specific regulations and is supervised by the ONS Secure Research Service (SRS) team. We will follow their guidance on statistical disclosure control and consult with the assigned project officer regarding what is deemed acceptable for publication. If no issues arise in terms of identifiability, we will include a breakdown of the ethnic composition of the sample in the participant section. However, to limit the number of predictors in the model, these categories will be recoded into the proposed binary variable for analysis where 0 = White and 1 = Non-white ethnic minority for regression analyses.

The impact of the COVID-19 pandemic on adolescent mental health: a natural experiment

Reviewer 2	
The general hypothesis is clear, plausible, and rationale (that Covid-19 impacted mental health). I would like to see the authors detail more explicitly what they think they will see in their analyses. Introduction. I would like to see the authors state their hypotheses clearly. What do they think the most likely outcome will be?	Given that results from existing literature are mixed and few studies account for pre-pandemic mental health and possible developmental trends, we originally chose not to specify a direction of the effect. On further consideration of this feedback and the existing literature, albeit methodologically limited, we have hypothesised that after controlling for baseline variables, levels in depressive symptoms and externalising difficulties will be higher, and life satisfaction lower, during the COVID-19 pandemic compared to before. Please see the inclusion of our hypotheses in the introduction (section 2.2, page 2), and in the study design table now included in the method section (page 10, Table 2).
I would have loved to see a full working analysis pipeline with simulated data, though I understand that is not always feasible.	We thank the reviewer for this suggestion. We have considered this as it is suggested in the submission guidelines as an option, but since the analyses follow clearly specified trial protocols (Hayes et al., 2019a 2019b) and use a standard analytical model, we did not see a particular point where this would help to further understand the procedure (especially with further defining the statistical model, see responses to reviewer 3 below). If there are points that remain unclear or open our plan up to unwanted flexibility, we would be happy to address them.
Mixed effect models. Please specify which variables will be used to define random intercepts. Presumably these will be student ID and school ID (though this is not specified anywhere). I think these should be nested (eg. (1 School/Student), though I could see an argument for them being crossed instead. Which will it be?	As the analysis outcome variable is mental health at follow-up (i.e., only one observation per participant), no random effect on student level will be specified. In addition to the full specification in Table 1, we have also added to the description of the primary outcome analysis in section 3.5.2. page 5 “with depressive symptoms at 1-year follow-up as the dependent variable.” We have also included the regression equations in section 3.5.2 to indicate our model specification.

The impact of the COVID-19 pandemic on adolescent mental health: a natural experiment

	We have additionally clarified on page 5 "The primary outcome analysis will use a random intercept (for schools) linear multivariable regression model..."
Covariates. How will Geographical region be coded? Will it be a dummy-coded variable? Or will it also be a random intercept?	We thank the reviewer for querying this and have now further clarified the role of geographical region in the analysis. Region will not be a random intercept and it was only included as a covariate as it was used for minimisation in the original trial. We have revisited this and think these have no bearing on this analysis. We will dummy code region and provide descriptive information on regional spread in the descriptive section. We have now clarified this in Methods sections 3.1, 3.3.2 and 3.5.1.
Interactions. a. Will variables be centered prior to forming the interaction term? (Presumably yes, but this should be explicitly stated).	All modifiers of interest in the study (e.g., sex, ethnicity, free school meal eligibility and, SEN) are binary categorical variables. Although we acknowledge that we could centre these binary categorical variables, given that the interaction terms are directly interpretable if variables are not centered it is not necessary in this analysis. Table 1 shows our decision to centre the baseline mental health outcome score. We have also made a note of this in the analysis plan (page 6, section 3.5.4). "Given the modifiers of interest are all binary categorical (coded 0,1) these will be entered into the models as is, as these interaction terms remain directly interpretable."
4. Interactions. b. Will the authors perform follow-up tests to aid in interpreting the interactions (e.g. Johnson-Neyman/Region of Significance, proportion of interaction, or cross-point)?	Thank you for this suggestion. Given that our interaction analyses are exploratory, and we report coefficients for all models and visual displays of those where a potential interaction might be of interest, readers will have all the information they need to cautiously interpret the results (or aggregate for meta-analyses). We would not want to put undue emphasis on these results.

The impact of the COVID-19 pandemic on adolescent mental health: a natural experiment

Sensitivity analyses. I would like to see additional sensitivity analyses for interaction analyses, in which covariate x main variable interaction terms are additionally included (to control for potentially spurious interactions, see https://doi.org/10.1016/j.biopsycho.2013.09.006). For example, if the main interaction analysis is: Depression = Covid x Gender + Covid + Gender + Age + SES etc... The sensitivity analysis would look like: Depression = Covid x Gender + Covid + Gender + Age + SES + Covid x Age + Covid x SES + Gender x Age + Gender x SES etc...	We thank the reviewer for this suggestion and are happy to include the suggested model as a sensitivity analysis of the interaction analyses. We have now included this information on page 6 in section 3.5.4. “A sensitivity analysis for these models will be conducted where all modifiers and their interaction with phase are included in the same model”
Alpha & multiple test correction. a. Alpha appears set to 0.05 for the main effect analyses (depression, life satisfaction, extraversion), but 0.10 for the interaction analyses. Why this difference? The authors state they will plot interactions that are $p < 0.10$. What interpretation will they ascribe to these interactions? That they are significant?	Given the large power required for interaction effects, we have used a less conservative threshold as an indicator of which modification analysis to further visualise in plots. We will not refer to these as ‘significant’ nor will attribute causal impacts to these findings. These plots will only be used to illustrate potential sub-group differences and will be discussed as exploratory findings.
Alpha & multiple test correction. b. Additionally, given that the authors appear to planning on performing 15 tests in total (1 primary+ 2 secondary + 4 modifiers x 3 main effects), I think it is worth raising the question of multiple test correction, which is not addressed anywhere. With $\alpha = .1$, at least one of the interaction tests will be “significant”, though I am not convinced that it will be meaningful. Are the authors planning on correcting for multiple comparisons? I am less concerned about the primary and secondary outcomes, as the authors have clearly stated how they will interpret these, as well as which is “primary”.	We are investigating one main hypothesis with one primary (and two secondary) outcomes. Everything else we report is exploratory and a p threshold is only used for those analyses as a conventional threshold to identify potentially relevant signal to noise ratios but won't be used to answer our main research question (i.e., not to make any claims about the potential causal impacts). Therefore, we do not think that multiple test correction is necessary. The separation into primary and secondary outcomes was introduced in section 2.2- Objectives; and the exploratory modification analyses are clearly indicated in the methods section. We have now also made this clear in the new PCIRR table included (Table 2).
Effect sizes. In the power analyses, please specify what kind of effect the “MDES” is (e.g. standardized regression coefficient).	Previous studies in the field use the Minimally Detectable Effect Size which is generally defined as the difference in averages between

The impact of the COVID-19 pandemic on adolescent mental health: a natural experiment

	two groups divided by the standard deviation of the focal variable (see also reference 29 that is provided). We have added this in parenthesis next to MDES to further make this clear in section 3.5.6 page 6.
Limitations. This study will give some indication of whether Covid-19 causally impacted adolescent’s mental health in the UK. Can the authors lay out the limitations of this study and reasons why we will not be able to fully ascribe causation if their results are significant?	We have now included a list of possible reasons that will limit our ability to fully ascribe causality in the limitations section on page 7 as follows: “The following will limit our ability to fully ascribe causation to our study findings (whether they support the hypothesis or not). First, the student composition of the two study phases might already differ at baseline. Second, differential attrition across the two phases: the response rates at follow-up for the two phases are unlikely to be the same, and the predictors of non-response might vary in the pre-pandemic and COVID-19 phases of the study. Third, there might be imbalances in the distribution of the interventions and controls across the two phases, and it is also plausible that differential effectiveness of the interventions across COVID-19 and pre-pandemic phases of the study might impact on our current analysis in unforeseeable ways. Finally, the pandemic experience could have had an impact on how students interpret or respond to the outcome measures.”
Reviewer 3	
This is Argyris Stringaris commenting on The impact of the COVID-19 pandemic on adolescent mental health: a natural experiment. This is a very worthwhile and thoughtful use of a natural experiment--two samples that should be comparable in principle completed close to each other, but one before the pandemic, the other with this second wave of data collection, within the pandemic.	Thank you for this comment and the positive feedback.
It was hard for me to follow the details of the mixed effects model. I am not sure how the random effects are specified. To me there seem	As described in the response to Reviewer 2 above, the model does not contain multiple observations per student. Table 1 sets out all

The impact of the COVID-19 pandemic on adolescent mental health: a natural experiment

to be too: one random effect is the schools, the other is individuals over time. Do the authors take both into account? Also, it would be good to specify clearly and unambiguously the covariates of interest in advance. Ideally, the complete regression equation should be presented here. A maximal random effects structure specification should be preferred and the criteria for fit/convergence declared (see Barr et al 2013 J Mem Lang, Bell et al Qual and Quand 2019).	the covariates that will be used in the model as independent variables. These are the variables collected at Baseline (Figure 2). The dependent variable is the score at 1-year Follow-up. In response to Reviewer 2 we have made the following changes: In addition to the full specification in Table 1, we have also added to the description of the primary outcome analysis in section 3.5.2. page 5 “with depressive symptoms at 1-year follow-up as the dependent variable.” We have also included the regression equations in section 3.5.2 to indicate our model specification. We have additionally clarified on page 5 "The primary outcome analysis will use a random intercept (for schools) linear multivariable regression model..." Since the model assumes only one random intercept on school-level, no additional convergence criteria are defined as these models can robustly estimate under a range of conditions.
My other recommendation would be to specify some effect size of interest. The p-value is a less meaningful here, particularly given the expected sample size. I believe the authors should specify some minimal effect of interest (there is guidance for this for the MFQs (eg Liu et al 2019 JAACAP). How many points of a difference would seem important?	We thank the reviewer for this suggestion. As per the paper directed to, 6 points change on the SMFQ has been suggested for clinical populations receiving treatment for depression (Liu & Adrian, 2019, JAACAP). Similar estimates for population-based samples are not already established and applying a RCI score derived from clinical samples (who tend to have high scores with more scope for reduction), would be inappropriate in our community-based sample. Even small effect sizes of change over a large population can have important implications for population health (e.g., if population average BMI changed by 1 point) and these can also be conceptualised in distributional terms (e.g., X% of a standard deviation) or standardised mean differences. We will

The impact of the COVID-19 pandemic on adolescent mental health: a natural experiment

	present distributional effect sizes in terms of standard deviation units of our outcomes (similar to the Cohen's d). This information has now been added to the methods section 3.5.2. We posit that anything over a 10% of a SD shift would be a meaningful effect size at the population level. “While it is difficult to provide a general cut-off for what is a relevant effect size in population-based research, in this study anything above a 10% of a standard deviation change in continuous scores would be considered an effect with potentially practical significance at population level (29).” An additional benefit of doing this for our primary and secondary outcomes is that it will also allow us to compare effect sizes across outcomes (which is currently not possible given the different scales of the measures). An additional effect size estimate, as recommended for population-based research (see Mathay et al. 2021, SSM Pop Health), is the population attributable fraction (PAF; Mansournia & Altman, 2018, BMJ). This allows one to estimate the number of cases that are attributable to the exposure of interest (i.e., COVID-19) and hence the proportion of cases fewer than might be expected in the absence of this exposure. We believe this practical measure of effect size will be a useful one for this study and have included this in the Methods section 3.5.2.
I love the authors' sensitivity analyses plans, all three of them are very appropriate. Again though, I would like to have some a priori notion of when the samples would be deemed comparable.	We appreciate the reviewer's positive feedback, and the point raised. Nevertheless, it is difficult to define clear cut-offs for the interpretation of propensity scores. We therefore further emphasised in our protocol that we will report the descriptive results of these analyses in detail (section 3.5.3, pages 5-6). “To describe the multivariate comparability of the two samples considering school- and individual-level variables, a propensity score will be estimated with a random intercept

The impact of the COVID-19 pandemic on adolescent mental health: a natural experiment

	logistic regression model, first for individual level only and then for a model with both school- and individual-level baseline variables as predictors of “phase”. We will separately visualise the distribution of the two propensity scores across the cohorts; and we will use the Stata module psmatch2 (-pstest-; 32) and 1-to-1 matching to report descriptive statistics for the included covariates (see Table 1), with and without matching. Although this will result in some unmatched students across phases (due to unequal sample sizes), this descriptive method offers comprehensive insight into the comparability of the underlying samples with respect to the available characteristics.”
--	---

Appendix B

Dear Prof Chris Chambers,

We would like to submit **stage 2** of our *registered report* (*stage 1 was in principle accepted in Aug 2021*) titled ‘**The impact of the COVID-19 pandemic on adolescent mental health: a natural experiment**’ to be considered for publication in *Royal Society Open Science*.

Responses to your queries raised on 17th Feb 2022 are below:

"1. Section 3.57 reports a major deviation in the original analysis plan. As noted in the Stage 1 IPA decision letter, any deviations from the approved study procedures must be communicated to the editor as soon as the authors realise it is necessary so that it can be properly assessed (and if need be subjected to re-review). Unless I've misplaced the email records, I don't think the authors consulted with the journal in this case. Please state in section 3.57 at which point in the research process this change in analysis was decided – i.e. prior to data observation, after data observation but prior to attempting the data analysis, or after the data analysis had been attempted and outcomes were known, so that we can assess risk of bias. As noted prominently in the RSOS policy, deviations from protocol that take place without prior editorial approval, and which increase risk of bias, can result in Stage 2 rejection.

Response: We report 3 deviations here in section 3.5.7.

1) the MLM is not possible given schools belong to either one phase and hence these models are not suitable, and we only realised this when we went to run them;

2) Invariance analysis, this is additional analysis reporting on measurement properties of the measures that we thought would be useful. We highlighted the lack of invariance as a possible limitation in the RR, and these measurement analyses help inform this consideration;

3) Prior mental health as a stratifier; this is an additional analysis that was suggested to us after the publication of the RR prior to any analyses taking place; we accept we should have written to ask for approval and did not realise we should have done this. However, we make it super clear that this is exploratory and we do not discuss the findings in any detail (other than to state them) or pay any undue attention to them in our discussions and implications. It would have been a missed opportunity to not include this after it was pointed out to us as being of relevance after the publication of the RR.

Deviations 1 and 2 are methodological and do not add any research questions or hypothesis. The only substantive deviation is the third one. We sincerely hope the fact that we report it transparently as a deviation and do not focus unduly on these results will be considered before rejecting our paper at this stage.

2. To assist the reviewers in comparing the Stage 1 manuscript with the final paper, please include a tracked-changes version of the Stage 2 manuscript that shows all text changes (however minor) between the Stage 1 manuscript and the corresponding sections of the Stage 2 manuscript (up to the Results section).

Response: Apologies, this had been created but we were unable to upload it previously as it replaces the manuscript designated document and there was no suitable option to upload it

under. We have uploaded it now under the reviewer responses designation (hope that is okay).

3. In the Results section of the Abstract, make clear which of the reported outcomes are confirmatory and which are exploratory. Same for the Conclusions section – and ensure that the conclusions are weighted toward the confirmatory outcomes.

Response: This has now been done. The discussion and conclusions are mainly weighted towards the confirmatory ones.

4. In the Results section of the main manuscript, please make clear either in the subheading titles, or in the definitions of individual analyses within sub-sections, which analyses are preregistered/confirmatory, and which are post-hoc/exploratory."

Response: We have now clearly indicated which analyses are not pre-registered by using a subheading.

We look forward to receiving reviewer feedback and your decision.

Yours faithfully,

Rosie Mansfield, Joao Santos, Jessica Deighton, Tjasa Velikonja, Jan Boehnke and Praveetha Patalay

Correspondence to: Praveetha Patalay, University College London

Email: p.patalay@ucl.ac.uk

Appendix C

Associate Editor Comments to Author (Professor Chris Chambers):

Associate Editor: 1

Comments to the Author:

One of the original Stage 1 reviewers was available to assess the Stage 2 manuscript, and I have decided that the reviewer's evaluation, combined with my own reading, is sufficient for us to continue with an interim editorial decision. As you will see, the reviewer broadly judges the Stage 2 criteria to be met, with the exception that conclusions based on exploratory analyses should be appropriately restrained. This comment is in line with the discussion concerning outcome reporting that we had at the pre-review stage, so please attend to it carefully.

RESPONSE: We have edited the manuscript further to indicate conclusions based on exploratory analysis are restrained as suggested by the reviewer.

Concerning the issue of data availability, this doesn't fall under Stage 2 criterion 1 (as implied by the reviewer's comment), but the reviewer is correct that this is a more general requirement of publishing in RSOS, unless special circumstances apply. The RSOS admin team will liaise with you separately to ensure these requirements are met.

RESPONSE: We had discussed the issues with the data availability prior to Stage 1 submission with the editorial team given the ongoing nature of the trials from which these data are analysed. As agreed with the editorial team (correspondence with Andrew Dunn, Senior publishing editor in July 2021) we said we would make the code available at Stage 2 alongside the paper with the data becoming available on OSF once the main trial results were published. This was agreed with the journal prior to Stage 1 review and the circumstances and constraints have not changed since then. This is also clearly communicated in the Data Accessibility Statement of the paper (copied below).

Data Accessibility

As per our trial protocols (21, 22), the raw data can only be shared after the trials are complete. An anonymised quantitative dataset was originally due to be made available in 2022 once the study had finished, however, due to disruption caused by the COVID-19 pandemic, this timeline has been delayed. Analysis code is shared as a supplementary file of this paper and on the Open Science Framework alongside the full stage 2 paper and the data will be made available on this same OSF project location subsequently after the main trial results are published.

Comments to Author:

Reviewer: 2

Comments to the Author(s)

1) Whether the data are able to test the authors' proposed hypotheses by passing the approved outcome-neutral criteria (such as absence of floor and ceiling effects or success of positive controls)

- No - data are not available, though it is stated that they are. I am recommending a 'major' revision for this reason, as data availability is a condition of publication at this journal.

RESPONSE: We do not state that the data are available and have clearly stated why in the data availability statement (see above).

2) Whether the Introduction, rationale and stated hypotheses are the same as the approved Stage 1 submission

- Yes

3) Whether the authors adhered precisely to the registered experimental procedures

- Yes

4) Where applicable, whether any unregistered exploratory statistical analyses are justified, methodologically sound, and informative

- Yes

5) Whether the authors' conclusions are justified given the data

- Conclusions regarding the moderating effect of sex (an exploratory analysis without multiple testing correction) need to be tempered, particularly in the abstract (which is what most people will read). For example:

Exploratory analyses ... *suggest* that the negative impact of ...

Females *may have been* impacted by the pandemic more than males across *most* outcomes

RESPONSE: These suggestions have been incorporated and results from exploratory subgroup analyses have been further tempered in both the abstract and the discussion.